# The Impact of COVID-19 and Its Policy Responses on Local Economy and Health Conditions

**Ali Gungoraydinoglu [1], Ilke Öztekin [2] and Özde Öztekin [1],***

1   College of Business, Florida International University, 11200 S.W. 8th St., Miami, FL 33199, USA; agungora@fiu.edu
2   College of Arts, Sciences & Education, Florida International University, 11200 S.W. 8th St., Miami, FL 33199, USA; igillam@fiu.edu
*   Correspondence: ooztekin@fiu.edu

**Abstract:** US states have implemented lockdown measures to contain the COVID-19 pandemic. We assess the impact of state policy responses on local economic and health conditions, with the goal to shed light on marginal health benefits and economic costs associated with social distancing. We find that lockdown measures are effective in alleviating disease severity, but yield significant contraction of the economy. Deteriorating health conditions are disruptive to the labor supply, financial health, and economic output. The adverse economic impact of lockdowns exceeds the economic damage brought by the disease itself, but health conditions better forecast economic contraction outcomes.

**Keywords:** COVID-19; pandemic; lockdown; bankruptcy; unemployment; economic growth

## 1. Introduction

The world is confronting a joint health and economic crisis of unprecedented severity. On 11 March 2020, the World Health Organization (WHO) declared the highly infectious coronavirus disease (COVID-19) to be a pandemic. Nations around the world have adapted measures of social distancing, in an effort to slow down the spread of the virus and save lives (Briscese et al. 2020; Merelli 2020; Paun et al. 2020). Some of the measures implemented include stay home orders, work and school closures, travel and social gathering bans, and postponement of the primary elections. Pandemic events present us with a natural experiment to test the relationship between social actions and the capacity for public and private response, as well as allow causal inferences to be made on the impact of health conditions and disease environments on economic outcomes.

Did policy actions to contain the spread of the COVID-19 disease save and sustain lives? Did such mitigation interventions have detrimental effects on the economic conditions of an average citizen and firm? Although these questions are the focus of attention for communities, public health experts, and policymakers, to our knowledge, there is no academic research directly linking the joint impact of public-health measures and disease severity to various local economic and health outcomes. The goal of the current investigation is to provide such evidence, based on US state-level policy actions.

In the face of serious health, as well as financial and real economic risks, national and local governments have a critical role. The mode and extent to which measures were implemented to contain the outbreak vary widely across regions. Countries around the world and states within the United States differed in their outbreak management strategies, as well as in the timelines of the interventions they chose to employ. The spread of the disease and its ultimate health and economic burden is a product of the decisions made by people and the conditions that lie behind these decisions. As such, the social distancing policies are purposefully inducing an economic slowdown. While preventive action on

pandemics is key to alleviate illness, death, and disruptions to public health, the evidence of the efficiency and economic side-effects of such actions is not well understood. This paper focuses on the management of infectious disease within the individual states of the United States and evaluates whether the heterogeneity in state-wide policymaking helps explain variations in local health and economic outcomes.

An important dimension of epidemics is their demographic consequences. Epidemics are a negative shock to the population. While illnesses suffered lead to absenteeism and lower productivity, premature deaths directly reduce the labor force (e.g., Bartel and Taubman 1979; Fan et al. 2016). Another important dimension of epidemics is their adverse effects on economic activity through the destruction of human capital, as well as through the individual and social measures intended to reduce the spread of the virus. For example, Fernandes (2020) estimates that GDP growth would decline by 3–5% in a mild scenario depending on the country, with a cost of about 2–2.5% of global GDP growth for each additional month of a shutdown. COVID-19 influences the local economy in two ways (Fornaro and Wolf 2020). First, the fear of the disease leads to a substantial decline in consumer demand. Second, disruptions in the labor market and the supply chain result in negative supply shocks to the economy. The more persistent the health shock is, the larger these negative influences are. An important role of governments is to manage this risk and coordinate policy responses locally, nationally, and globally. A global pandemic shock crucially requires action to minimize morbidity (unable to work when incapacitated or caring for the incapacitated due to the illness) and mortality (deaths). However, it is also important to attend to the detrimental influences of the pandemic on the economic, social, and political stability of communities. As such, the lessons we have learned from a comparative approach towards the major containment measures implemented within the United States may help policy makers modify outbreak management strategies for better health and economic outcomes in the future.

Thus far, the effects of state policy actions on public health as well as the real economy and the welfare of average Americans and businesses have not been directly assessed. To assess the effects of policy actions meant to contain the coronavirus disease on the local market economic and health conditions, we evaluated the related changes in local economic and health conditions as a function of the relative strictness of state policy measures and disease severity in the local areas. For instance, if state policy actions are effective in saving lives, then one would expect to see the health conditions in the states that adopted more strict containment measures deteriorate less compared to states that adapted less restrictive policy measures. Moreover, if there is a notable health versus economic tradeoff, the real economic burden of such policy actions would be more apparent within states that undertook more severe lockdown measures. In this paper, we leverage several indicators of local economic conditions that likely affect average Americans and businesses: real economic growth, state coincident index, business revenues and business employment, state unemployment, and the total number of personal and business bankruptcies, treating each state as the key local area. Our target explanatory variables include various measures of the restrictiveness of the social distancing measures imposed by each state and the severity of the disease as measured by illness incidences and deaths.

This paper uses the coronavirus pandemic as a tool to assess the causal effect of health conditions and state policy responses on the local economy. The episode of large and sudden exogenous increases in coronavirus illness cases and fatalities helps us isolate exogenous variations in health conditions across different states within the United States. Our findings indicate that both health conditions and lockdown measures are important for the real economy. However, the direct impact of disease environments on economic growth is less apparent in the short term, whereas lockdown measures have strong adverse effects in the near term. Specifically, health conditions as captured by the changes in cases and deaths from the coronavirus disease have a significant negative impact on economic outcomes but are unlikely to be the major reason for the large immediate losses in the economic output. Instead, the heterogeneity in policymaker responses to the health crisis

and the severity of the state lockdowns appear more likely to be the main factor behind these income differences. Disease severity, on the other hand, had a more profound effect in increasing state unemployment and the number of personal and business bankruptcies. Our analyses also shed light on the relative importance of state policy actions and disease severity in forecasting contractions in real economic activity. Specifically, a machine learning assessment of the relative strength of coronavirus disease severity and state policy measures to predict real local economic contraction outcomes indicates that the disease itself has more predictive utility relative to mandatory social distancing policy measures undertaken by the states to slow down the spread of the disease.

Our findings provide further insight into several important aspects of disease management during the pandemic. First, an overview of the determinants of the state policy actions undertaken to mitigate the spread of the disease and its severity suggests that states with higher death incidences have imposed stricter restrictions broadly and stay home orders, business closures, and primary election postponement more narrowly. While deaths are an important determinant of the severity of state actions, with the exception of election postponement, cases alone do not seem to affect the restrictiveness of the state mitigation measures.

Second, our findings indicate the main channel of impact for the health and economic consequences of state policy actions to be through restricting individual behavior and reducing mobility. That is, mobility shows a significant downward trend in states with stricter mitigation policies using the overall state policy restriction measure as well as its stay home and business closure components. Three policy measures—traveler quarantines, gathering bans, and primary election postponement—are similarly associated with declines in mobility, but these effects were not statistically significant. Our results also confirm the major benefit of disease testing in reducing the spread of the disease. Higher testing rates lead to significant reductions in mobility and this deterring effect supersedes the impact of state policy actions when gathering bans and primary election postponement are considered as the social distancing measures.

Third, health gains from stricter policy actions come primarily through reducing incidences of disease cases rather than disease fatalities. Furthermore, not all state policy measures yielded similar outcomes. Among the social distancing measures, state restrictions, stay home orders, business closures, and gathering bans are identified to be the only effective mitigating interventions that helped slow down disease cases. As for the incidences of disease deaths, the only state policy measure that was identified to be essential was gathering bans. Traveler quarantines and primary election postponement policy measures, on the other hand, did not seem to make an important difference in improving health outcomes, at least in the short term.

Fourth, with respect to the effects of stricter state restrictions and higher disease morbidity and mortality on the local economy, we assess both their quantity (the severity of the economic damage) and quality (identification of a real economic contraction outcome) implications. With respect to the relative severity of the disease and social policy outcomes, we show that disease fatalities have a more notable negative economic impact compared to disease cases, but the economic impact of health conditions is relatively smaller than the economic effects of state policy measures. For example, state restrictions result in a 0.46 standard deviation drop in real economic output, whereas the corresponding decreases are 0.37 and 0.40 standard deviations with disease cases and deaths, respectively. With regard to the identification of real economic contraction outcomes within US states, we utilize machine learning and assess the predictive utility of our social policy and disease severity variables in forecasting local real economic contraction. Our findings reveal that the highest explanatory power in predicting real economic conditions arises from state restrictions, stay home orders, business closures, as well as disease cases and deaths. When a horse race is run between state policy measures and disease severity, with the exception of state restrictions and business closures, disease severity outperforms social distancing measures in predicting real economic conditions in most specifications. Thus, rather than the state-

mandated social distancing measures, the disease itself is a crucial factor in forecasting disease-induced real local economic contraction outcomes within the US states.

Finally, we evaluate the specific channels through which negative economic outcomes occur either due to disease severity and social policy. The impact of disease severity on the real economy comes primarily through increasing state unemployment and the number of personal and business bankruptcies. Social distancing measures, on the other hand, are significant determinants of the general economic activity level as measured by the state coincident index, as well as business earnings and employment within the US states.

Below, we describe in detail the state policy measures assessed (Section 2) and review the relevant literature (Section 3). Section 4 describes the methods, the sample, and the data evaluated. Section 5 presents model specifications and the results. The conclusion of our paper is presented in Section 6.

## 2. Brief Description of the State Policy Actions to Mitigate the Spread of COVID-19

Within two days of the coronavirus virus being declared as a pandemic by the World Health Organization, the federal government of the United States similarly declared a state of emergency. By March 16, every US state had made an emergency declaration. The states responded to the rapidly growing disease caused by the new strain of the coronavirus with a number of actions intended to mitigate the spread of the virus. An important policy initiative was to encourage critical spatial behaviors to help contain the transmission of the disease. According to Hafiz et al. (2020), three important, interrelated policy concerns demand attention from regulators: social insurance, broader economic and systemic management, and spatio-behavioral management. In this paper, our focus was on the third component, namely lockdowns, and physical distancing measures. The states that had a more prominent outbreak of the disease took more severe and timely actions to slow the contagion in their local area. This study investigates the initial round of social distancing measures that included mandatory stay-at-home orders, closures of non-essential businesses, bans of large gatherings, school closures, limits on bars and restaurants, and other public places, as well as primary election postponement. We excluded two measures (school closures and the declaration of emergency) from our analyses as all states uniformly implemented these measures. We also acknowledge that after having social distancing requirements in place for several weeks, states began to ease back some of the measures, allowing non-essential businesses to open, rescinding stay-at-home orders, easing restrictions on in-person dining at restaurants, and easing gathering bans. As of May and June 2020, many states started to loosen social distancing restrictions in some way. Our main focus remains on the initial restrictive state actions that were in effect during the first quarter of 2020. Thus, it is important to note that our baseline analyses did not consider the impact of the relaxation of these constraints over time.

Global and local leaders bear in mind the impact of such measures on their economy, and the extent to which the chosen policy measures have been effective in slowing down the growth of new infections and fatalities remains an important research question. The federal structure of the United States and lack of national policy coordination led to significant variations in state policies and raised questions regarding the degree of effectiveness of the taken measures for containing the virus while minimizing the resultant burden on the economy. The differences between states in the approaches to disease management and the systemic evaluation of their consequences are particularly crucial for gaining insight on the relative efficiency of alternative responses to the health crisis and the appropriate timing for rolling back such restrictions. We evaluated which measures were more effective in reducing social mobility and slowing down the spread of the virus while minimizing the contraction in the economic activity level. In doing so, our analyses enabled systemic cross-state comparisons of government interventions and their economic and health impact.

## 3. Related Literature and Contributions

Health conditions and disease environment are important for economic outcomes (Acemoglu et al. 2001, 2002, 2003). Numerous studies have documented that population health, as measured by life expectancy or mortality, is positively related to economic welfare (e.g., Pritchett and Summers 1996; Bloom et al. 1998; Bhargava et al. 2001; Cuddington et al. 1994; Cuddington and Hancock 1994; Robalino et al. 2002a, 2002b; Haacker 2004). Is it useful for states to intervene with the pandemic? If so, what policy measures are proven to be useful? While improvements in health may positively contribute to economic growth, increased mortality may adversely affect long-term productivity. Human capital plays a key role in economics: "net of medical expenditures, the value of increased life expectancy between 1970 and 2000 in the United States" equaled "a flow of about $2 trillion per year" (Murphy and Topel 2006).

Mushkin (1962) emphasizes several important aspects of public health: (i) health is complementary to other forms of human capital, particularly education, (ii) health-related changes have an important but hard to measure impact on the quality of life, (iii) health contributes to increasing longevity and economic growth, (iv) health research and treatment are public goods requiring government intervention, (v) health improvements have monetary value. Improved health is not only valuable in terms of increased labor supply and production derived from productivity and the ability to work, but also for its diverse benefits arising from living a longer and better life. In short, health is inevitably valuable for all, and the rationale for government interventions to contain the spread of the infectious disease is very straightforward: flattening the epidemiological curve helps avoid bottlenecks in the healthcare system that result in suboptimal treatment and thus more avoidable deaths. At the same time, keeping workers away from work and consumers away from consumption may inevitably reduce economic activity and welfare.

In addition to its detrimental impact on public health, the coronavirus outbreak has also caused a significant negative demand and supply shock to the economy (Fornaro and Wolf 2020; Eichenbaum et al. 2020; Guerrieri et al. 2020; Gourinchas 2020; Petrosky-Nadeau and Valletta 2020). On the supply side, the direct effects of the pandemic health shock involve a direct contraction in the supply of labor from unwell workers, caregivers who stay home to take care of children during school closures, and increased mortality. The indirect, maybe even larger, impact on economic activity arises from the efforts to contain the spread of the disease through lockdowns that inevitably lead to the underutilization of the productive capacity. On the demand side, the loss of income, fear of disease spread, and heightened uncertainty lead people to spend less, making firms unable to pay worker salaries, resulting in layoffs, further exacerbating business closures, job and output losses.

Recent research (e.g., Jordà et al. 2020) evaluated the long-term macroeconomic effects of global pandemics. Here we focus on how the COVID-19 pandemic impacted the US local economy and labor market in the initial few weeks of the health crisis. Specifically, we evaluate the economic burden of the state actions taken in response to the disease outbreak. Social distancing measures required closing schools and non-essential businesses, requiring most of the working-age population to stay at home. It is thus important to quantify the extent to which the lockdowns aimed at containing the virus led to decreases in economic output.

Our findings complement the growing literature on the pandemic economy. Several researchers studied the impact of COVID-19 on corporate firms and financial markets and institutions. For instance, Beck (2020) focuses on finance and banking risks, Li et al. (2020) study banks as liquidity providers to nonfinancial businesses, Carletti et al. (2020) forecast the drop in profits and the equity shortfall due to COVID-19 lockdowns for a sample of Italian firms, Acharya and Steffen (2020) assess the relation between COVID-19 induced credit risk and corporate cash holdings, Ramelli and Wagner (2020) document market reactions to COVID-19, Halling et al. (2020) demonstrate the patterns of public capital market access for a sample of US firms, and Baek et al. (2020) evaluate unemployment effects of lockdowns. While not directly evaluating COVID-19, other papers have evaluated similar past episodes

(e.g., Adda 2016; Barro et al. 2020; Markel et al. 2007; Correia et al. 2020; Velde 2020). In this paper, we evaluate the effects of two distinct COVID-19-induced economic shocks: a health shock as measured by the number of cases and deaths from the disease, which results in reduced labor and economic activity, and the economic impact of the disease containment measures implemented in individual states across the nation, which similarly reduces economic welfare. An infectious outbreak directly influences an economy by creating losses in the current and future income due to increased incidences of illness and death. In addition to the direct effect of the health risk associated with the pandemic, the states across the nation have quickly adapted several social distancing measures to quell the pandemic, with additional demand and supply shocks to the economy. To be able to quantify these effects, we assessed the joint impact of the health shock and state policy responses to it on state-level economic prospects.

Are health conditions an important determinant of individual and social welfare? Poor health may undermine the earning capacity of the labor force and the hours worked. Quantifying the productivity losses from poor health is challenging due to the difficulties in measuring the quality of health. An individual's self-evaluation of their health status is often a biased measure of health. Using the recent pandemic as an exogenous health shock, we estimated the real effects of health conditions. Specifically, we relied on active disease cases and deaths that affect the current earning power in the labor force. The results suggest that the disease had detrimental effects on individuals' labor market prospects, by reducing the labor supply.

Adverse health outcomes may not only trigger a persistent drop in labor productivity and directly limit employment, but could also generate significant reductions in the productive capacity of the local economy, by adversely affecting the labor market and the financial health of the affected companies. Since many occupations require in-person social interaction, the states have implemented economic lockdown measures that are aimed at mitigating the impact of the virus on public health. These shelter-in-place restrictions likely prevent active job search to various degrees depending on the severity of such measures enacted in the states. Our findings reveal that policy-induced social distancing practices play a significant role in disrupting labor productivity and economic activity, and thereby causing a massive reduction in the output of goods and services. Similarly, the coronavirus shock itself is leading to an economic contraction by inducing significant spikes in the level of local unemployment and the number of bankruptcies.

Overall, we show that the main impact of disease environments on the economy is both due to their direct effect as health conditions on income as well their indirect effect through state policy actions implemented to contain the disease. Health affects the economy directly by spiking unemployment and bankruptcies but explains a smaller fraction of the cross-sectional differences in state per capita income. State public health containment measures have a major adverse impact on the local economy with first-order consequences for economic activity and growth, by dampening business earnings and employment. Yet, the disease itself rather than the state-mandated social distancing measures is a crucial factor in forecasting disease-induced real local economic contraction outcomes within the US states.

## 4. Sample and Data

Our sample consists of cross-sectional observations of individual US states. Our main independent variables of interest are measures of COVID-19 severity as measured by case and death rates adjusted for the state population as well as state policy measures on social distancing. COVID-19 severity data are obtained from the Center for Systems Science and Engineering (CSSE) at Johns Hopkins University and reflect the information till the end of the first quarter of 2020. The data are publicly available at https://systems.jhu.edu/ (accessed on 4 September 2020). Information on state actions is compiled from a review of state executive orders, guidance documents, policy bulletins, and news releases and made publicly available by the Kaiser Family Foundation (KFF). The information reported

reflects the status of state social distancing requirements, including stay-at home-orders, closures of non-essential businesses, limits on large gatherings, limits on restaurants and recreation, and postponement of primary elections. The data and data sources for state policy actions to address coronavirus are available from the Kaiser Family Foundation (KFF): https://www.kff.org/report-section/state-data-and-policy-actions-to-address-coronavirus-sources/ (accessed on 4 September 2020).

Our main dependent variables are various measures of local health and economic conditions. For health outcome analyses, changes in COVID-19 cases and deaths are measured from the first quarter of 2020 to the second quarter of 2020. Using local economic conditions, the main outcome variables are real Gross Domestic Product (GDP), state coincident index based on four state-level indicators of economic conditions (nonfarm payroll employment, average hours worked in manufacturing, unemployment rate, real wages, and salary disbursements), unemployment rate based on unemployment insurance claims per 100 people in the labor force, bankruptcy filings by individuals and businesses, business earnings and employment for low-income workers. These variables were obtained at the end of the first quarter of 2020, relative to the beginning of the same quarter or the end of the previous quarter. The data come from various sources, including the Bureau of Economic Analysis, the Department of Labor, the American Bankruptcy Institute, the Federal Reserve Bank of Philadelphia, and The Opportunity Insights Economic Tracker (Chetty et al. 2020).

To evaluate the effect of state policies on individual behavior, we rely on Google Community Mobility Reports that show how visits to places, such as grocery stores and workplaces, are changing in each geographic region. The data are publicly available online at https://www.google.com/covid19/mobility/ (accessed on 4 September 2020). Retail and recreation mobility captures movement trends for places like restaurants, cafes, shopping centers, theme parks, museums, libraries, and movie theaters. Grocery and pharmacy mobility reflects movement trends for places like grocery markets, food warehouses, farmers markets, specialty food shops, drug stores, and pharmacies. Transit mobility measures movement trends in public transport hubs such as subway, bus, and train stations. Workplace mobility shows movement trends for places of work. In each case, we measure changes from the baseline to the end of the first quarter of 2020. The baseline corresponds to the five-week period from 3 January to 6 February 2020.

The empirical analyses control for a variety of health and economic conditions: pneumonia shot share for age 65 and over, hospital beds per population, lagged change in real GDP, and state population. These variables are based on a variety of sources including the Kaiser Family Foundation (KFF), the COVID Tracking Project, the Bureau of Economic Analysis, and the Census Bureau.

Table 1 shows summary statistics and Appendix A provides a description and sources of the variables used in the study. We report variables within five related groupings: state policy actions, local health conditions, local economic conditions, mobility trends, and control variables. In Panel A, the averages of individual components of social distancing restrictions are 0.86 for stay home orders, 0.33 for traveler quarantines, 0.81 for business closures, 0.72 for gathering bans, and 0.27 for postponement of primary elections. The overall measure of state social distancing restrictions as measured by the first principal component of the individual indices has an average of 0.61. In Panel B, local health conditions worsen, with both cases and death rates going up significantly (by 25 and 51 times, respectively) at the end of the first quarter of 2020 relative to the beginning of the same quarter. By construction, about half of the US states experienced high case and death rates above the median level. In Panel C, except bankruptcy filings, local economic conditions similarly deteriorate on all fronts during the first quarter of 2020 relative to the last quarter of 2020. The average change in real GDP is −5%. The state coincident index went down by 1% and business earnings decreased by 2% on average. Business employment for all firms similarly went down by 2%, whereas small businesses only experienced a decline of 1%. Unemployment increased by 3% on average, yet surprisingly

bankruptcy filings declined on average by 6%. Panel D shows that all mobility trends are downward on average, ranging from 15% to 47%. By construction, overall mobility is on average zero, with minimum and maximum changes of −6% and 4%, respectively. In Panel E, on average, 73% of the population aged 65 and over have ever had a pneumonia shot; on average there are 2.64 hospital beds available per population; the testing rate per population is 2% on average; real GDP increased on average by 2% from the third quarter of 2019 to the fourth quarter of 2019, and the average state population was 6.4 million in 2019.

**Table 1.** Summary statistics. The table provides summary statistics of the variables used in the study.

| Variables | Mean | Std. Dev. | Min | Max |
|---|---|---|---|---|
| Panel A. State policy actions | | | | |
| State restrictions | 0.61 | 0.21 | 0.00 | 1.00 |
| Stay home | 0.86 | 0.34 | 0.00 | 1.00 |
| Traveler quarantine | 0.33 | 0.43 | 0.00 | 1.00 |
| Business closures | 0.81 | 0.36 | 0.00 | 1.00 |
| Gathering bans | 0.72 | 0.27 | 0.00 | 1.00 |
| Postpone elections | 0.27 | 0.45 | 0.00 | 1.00 |
| Panel B. Local health conditions | | | | |
| Case rate per 100,000 people | 15.39 | 15.50 | 1.01 | 33.66 |
| Death rate per 100,000 people | 0.28 | 0.26 | 0.00 | 0.58 |
| High cases | 0.49 | 0.50 | 0.00 | 1.00 |
| High deaths | 0.49 | 0.50 | 0.00 | 1.00 |
| Change in cases (%) | 25.47 | 19.66 | 3.12 | 110.49 |
| Change in deaths (%) | 50.85 | 39.79 | 3.31 | 176.22 |
| Panel C. Local economic conditions | | | | |
| Change in real GDP (%) | −0.05 | 0.01 | −0.08 | −0.01 |
| Change in coincident index (%) | −0.01 | 0.02 | −0.06 | 0.05 |
| Change in business earnings (%) | −0.02 | 0.04 | −0.08 | 0.11 |
| Change in small business earnings (%) | −0.02 | 0.04 | −0.10 | 0.09 |
| Change in business employment (%) | −0.02 | 0.03 | −0.06 | 0.09 |
| Change in small business employment (%) | −0.01 | 0.03 | −0.09 | 0.06 |
| Change in unemployment (%) | 0.03 | 0.02 | 0.01 | 0.11 |
| Change in bankruptcy filings (%) | −0.06 | 0.06 | −0.21 | 0.10 |
| Panel D. Mobility trends | | | | |
| Change in retail and recreation mobility (%) | −0.40 | 0.07 | −0.61 | −0.26 |
| Change in groceries and pharmacy mobility (%) | −0.15 | 0.07 | −0.32 | −0.02 |
| Change in transit mobility (%) | −0.41 | 0.16 | −0.77 | −0.09 |
| Change in workplace mobility (%) | −0.47 | 0.07 | −0.69 | −0.33 |
| Change in overall mobility (%) | 0.00 | 1.00 | −0.06 | 0.04 |
| Panel E. Control variables | | | | |
| Pneumonia shot share for age 65 and over | 0.73 | 0.04 | 0.64 | 0.79 |
| Hospital beds per population | 2.64 | 0.75 | 1.65 | 4.76 |
| Tests per population | 0.02 | 0.02 | 0.00 | 0.13 |
| Lagged change in real GDP | 0.02 | 0.00 | −0.00 | 0.03 |
| State population (millions) | 6.44 | 7.36 | 0.58 | 39.51 |

## 5. Empirical Methodology, Hypotheses, and Results

To examine the determinants of state policy actions, their channel of impact, and the joint impact of exogenous health shocks and state policy actions on the local economy as well as subsequent public health outcomes in a multivariate setting, we estimated regressions with the dependent variables alternatively set to state policy actions, mobility trends, economic outcomes, and health outcomes. The main independent variables of interest in the regressions for the determinants of state policy action are binary variables that equal 1 for the states with above-median COVID-19 case or death rate per 100,000 people. The explanatory variables of interest in the mobility trend and local health outcome regressions are various state policy actions. Finally, the key independent variables in the regressions

for local economic outcomes are various state policy actions as well as COVID-19 case and fatality rates. To gauge the economic magnitude of the impact of state mitigation and disease severity measures on local economic and health outcomes, we reported the standardized coefficients that refer to how many standard deviations the dependent variable changes per one standard deviation increase in the independent variables.

Our regressions took advantage of state-based variation in the pandemic situation using measures of disease severity and allowed us to isolate differences between states that were more or less susceptible to the COVID-shock. In essence, the estimations are difference-in-difference (DiD) regressions with the caveat that all groups receive the treatment to some degree. However, states received the treatment to a varying extent. Thus, there is still a distinction between the treatment and the control, because while all states in the model are exposed to the treatment, some states (treated) are largely affected by the pandemic (higher disease severity), while others (control) are affected to a smaller extent (lower disease severity). We confirmed that treated and control groups were on parallel trends before the onset of the pandemic. Appendix B provides evidence on the validity of the parallel trends.

We employed machine learning for predicting categorical outcomes in our data, and to further assess the relative feature importance rankings for our target variables to derive their relative predictive utility. An important advantage of machine learning algorithms compared to classical methods is their ability to explain high dimensional patterns, accurately classify an extensive number of categories, and thereby make superior out-of-sample predictions.

A Support Vector Machine with a linear kernel was trained to distinguish the local economic contraction outcome (contraction present or contraction absent) across the states from the target features consisting of the various state restrictions imposed and the COVID-19 cases and deaths. Economic contraction is defined as a decline in real GDP of more than one standard deviation below its mean calculated over the previous four quarters. All models included our control variable, lagged change in real GDP, as before. We were mainly interested in the overall and relative predictive ability of state policy and disease severity measures. To achieve this goal, we employed the scikit-learn (https://scikit-learn.org/stable/ accessed on 4 September 2020) open-source machine learning platform for constructing our models. In order to evaluate feature importance from classifier coefficients, we adapted a Support Vector Machine (SVM) classifier with a linear kernel. We scaled our features using the StandardScaler function built in the sci-kit machine learning library. To account for the imbalanced distribution of our local economic contraction outcome, we used the Synthetic Minority Over-Sampling Technique (SMOTE) during model training. For model validation, we leveraged the built-in cross-validation function of the scikit-learn library. Performance was then evaluated with the commonly employed accuracy scores, as well as $F_1$ scores obtained across the five cross-validation indices for each model. $F_1$ scores are a classification performance metric that is calculated based on the precision (p) and recall (r), $F_1 = \frac{2pr}{p+r}$.

To evaluate feature importance rankings, we leveraged the magnitude of classifier coefficients across the target features in the model. Weights obtained from the resultant classifier coefficients of linear SVM classifiers can be utilized to infer feature importance rankings within the assessed model. Thus, the average classifier coefficients across the five cross-validation indices can provide an index for the feature importance rankings for models where multiple features are evaluated. Baseline accuracy and $F_1$ metrics were calculated by training a dummy classifier that always predicted the positive class (i.e., local economic contraction present).

*5.1. Determinants of State Policy Actions*

We started our empirical analysis with an overview of the determinants of the state policy actions to mitigate disease spread and severity. Our main hypothesis centers on the contention that states with a higher prevalence of the disease should be more likely to adopt stricter mitigation strategies to combat the disease. To this end, our main independent

variables are binary indicators that equal 1 for the states with above-median COVID-19 case and death rates per 100,000 people. Significantly positive coefficient estimates on high cases and/or high deaths would indicate support for our hypothesis that states with higher exposure to the disease would be inclined to implement more severe mitigation policies to slow down the disease contagion. Our regression analyses controlled for other potential determinants of state policy actions. We used pneumonia shot share for people aged 65 and over as our main demographic characteristic. We used hospital beds per population as a measure of the state's public health infrastructure. We also controlled for the lagged change in the real GDP to take into account the economic well-being of the state. Our dependent variables constitute state policy indices. The results for the determinants of state policy actions in the US states are presented in Table 2. Panels A and B show the impact of high case and death incidences on the strictness of state mitigation measures, respectively.

**Table 2.** The determinants of state policy actions.

| | (1) | (2) | (3) | (4) | (5) | (6) |
|---|---|---|---|---|---|---|
| **Dependent Variable** | **State Restrictions** | **Stay Home** | **Traveler Quarantine** | **Business Closures** | **Gathering Bans** | **Postpone Elections** |
| Panel A. COVID-19 case severity | | | | | | |
| High cases | 0.0511 | 0.1969 | −0.1635 | 0.1639 | −0.0387 | 0.3015 ** |
| | (0.704) | (0.123) | (0.272) | (0.250) | (0.793) | (0.032) |
| Pneumonia shot share over 65 | −1.1330 | −1.8751 | −0.3244 | −0.4832 | −0.9830 | −1.8668 |
| | (0.173) | (0.131) | (0.862) | (0.744) | (0.397) | (0.302) |
| Hospital beds per pop | −0.1231 *** | −0.1980 *** | −0.0879 | −0.1639 ** | −0.0644 | 0.0154 |
| | (0.004) | (0.002) | (0.350) | (0.031) | (0.270) | (0.864) |
| Lagged change in Real GDP | −0.0316 | 0.0150 | −0.0757 | −0.0596 | 0.0242 | −0.1451 |
| | (0.441) | (0.806) | (0.417) | (0.420) | (0.674) | (0.110) |
| Constant | 1.8077 *** | 2.6515 *** | 1.0186 | 1.6573 | 1.5679 * | 1.7466 |
| | (0.003) | (0.004) | (0.447) | (0.122) | (0.063) | (0.178) |
| R-squared | 0.217 | 0.313 | 0.054 | 0.133 | 0.058 | 0.185 |
| Panel B. COVID-19 death severity | | | | | | |
| High deaths | 0.3101 ** | 0.3272 *** | −0.1075 | 0.3848 *** | 0.1713 | 0.5325 *** |
| | (0.017) | (0.008) | (0.469) | (0.005) | (0.239) | (0.000) |
| Pneumonia shot share over 65 | −0.8310 | −1.5789 | −0.2613 | −0.0016 | −0.6895 | −1.1807 |
| | (0.289) | (0.183) | (0.890) | (0.999) | (0.548) | (0.458) |
| Hospital beds per pop | −0.1201 *** | −0.1942 *** | −0.0885 | −0.1585 ** | −0.0619 | 0.0240 |
| | (0.003) | (0.002) | (0.351) | (0.025) | (0.282) | (0.762) |
| Lagged change in Real GDP | −0.0341 | 0.0221 | −0.0889 | −0.0563 | 0.0182 | −0.1314 * |
| | (0.372) | (0.699) | (0.339) | (0.407) | (0.745) | (0.095) |
| Constant | 1.5317 *** | 2.3691 *** | 0.9765 | 1.2084 | 1.3042 | 1.0962 |
| | (0.008) | (0.007) | (0.474) | (0.228) | (0.118) | (0.339) |
| R-squared | 0.307 | 0.379 | 0.040 | 0.250 | 0.085 | 0.372 |

The table presents the determinants of state policy actions in the US states. High cases (deaths) is a binary variable that equals 1 for the states with above-median COVID-19 case (death) rates per 100,000 people. The *p*-values are reported beneath the coefficient estimates. *, **, and *** indicate significance at the 10%, 5%, and 1% levels, respectively.

An examination of Table 2 indicates strong support for the initial severity of COVID-19 deaths having an important role in the determination of the severity of state actions. Specifically, states with higher death incidences have more severe state restrictions, stay home orders, business closures, and higher chances of primary election postponement. While disease mortality is an important determinant of the severity of the state actions, with the exception of election postponement, cases alone do not seem to affect the severity of the state mitigation measures. These findings lend partial support (using COVID-19 deaths primarily) to our hypothesis that states with a higher prevalence of the disease will be more likely to adopt stricter mitigation strategies to combat the disease.

*5.2. Effectiveness of State Policy Actions and Testing in Reducing Mobility*

In order for the state mitigation interventions to be successful in reducing COVID-19 cases and deaths, they need to effectively restrict mobility. The state policy actions aim to directly reduce crowding and increase social distancing with the goal of reducing

transmission, delaying transmission, and reducing epidemic peak. Our first hypothesis of this subsection tests whether state restrictions achieve their goal of reducing mobility. We hypothesized that mobility would be lower in states with stricter mitigation interventions. Furthermore, significant negative effects of state policy actions on mobility would indicate support for our hypothesis that disease mitigation policies were effective in promoting social distancing.

In addition, there is a lot of emphasis among epidemiologists and public health experts on increased testing capacity as one of the most effective ways to combat the COVID-19 pandemic. A recent study by Acemoglu et al. (2020) developed a theoretical model to evaluate the effects of testing on infections to provide new insights on optimal testing strategies. They modeled social activity and voluntary distancing as a network information problem. Their finding confirms the major benefit of testing in reducing the spread of the disease when social activity levels are taken as given. However, they have also shown that, when equilibrium involves some groups choosing an intermediate level of social activity because of their fear of infection, greater testing leads to more social activity, and less voluntary social distancing, thereby increasing infections. Their findings are consistent with precautionary tools increasing risk-taking. In light of these important theoretical insights, we empirically assessed the impact of testing on mobility as well. Our second hypothesis of this subsection tests whether increased testing results in voluntarily reduced mobility (or an increase in mobility). Significantly negative relations between mobility measures and disease testing would lend support to the hypothesis that higher disease testing distorts social distancing incentives (Acemoglu et al. 2020).

To test these hypotheses, Table 3 presents results on the joint impact of state policy actions and COVID-19 testing on mobility trends in the US states. The dependent variables are the change in retail and recreation mobility, groceries and pharmacy mobility, transit mobility, work mobility, and overall mobility shown in Panels A–E, respectively. Overall mobility is the first principal component of the individual mobility categories.

Our findings suggest that stricter state restrictions lead to reliable declines in individual mobility categories, as well as overall mobility, consistent with our first hypothesis above. In all panels, mobility shows a significant downward trend in states with stricter mitigation policies using three out of six policy measures: state restrictions, stay home orders, and business closures. While three policy measures, namely traveler quarantines, gathering bans, and primary election postponement, are similarly associated with declines in mobility, these effects did not reach statistical significance. Our results also indicate that mandatory state restrictions of social distancing achieve their goal of reducing mobility, even after controlling for the impact of disease testing on voluntary mobility.

Higher testing rates lead to significant reductions in mobility and in some cases supersede the impact of state policy actions. For example, in all panels, testing has a significantly negative impact on mobility, whereas social policy action is not statistically significant using gathering bans and primary election postponement. We did not find evidence that increased testing is associated with a decrease in voluntary social distancing. In other words, there is no support for our second hypothesis that testing results in an increase in mobility. This finding is not surprising, however, since data availability did not allow us to model complex social interactions of individuals within states to incorporate the endogeneity of social network formation into our empirical analyses. Thus, we note that it is still possible that within states, there is a heterogeneous response of mobility to testing based on individual risk profiles. Our analyses were only able to capture the average effect.

**Table 3.** Effects of state policy policies and COVID-19 testing on mobility trends.

| State Policy | (1) State Restrictions | (2) Stay Home | (3) Traveler Quarantine | (4) Business Closures | (5) Gathering Bans | (6) Postpone Elections |
|---|---|---|---|---|---|---|
| **Panel A. State policy effects on changes in retail and recreation mobility** | | | | | | |
| State policy | −0.3477 *** | −0.3789 *** | −0.0370 | −0.3700 *** | −0.1154 | −0.1288 |
| | (0.007) | (0.003) | (0.788) | (0.004) | (0.398) | (0.335) |
| Tests per population | −0.3381 *** | −0.3374 *** | −0.3801 | −0.3469 *** | −0.3713 *** | −0.3972 *** |
| | (0.009) | (0.008) | (0.006) | (0.007) | (0.007) | (0.004) |
| State population | −0.0000 | −0.0000 | −0.0000 | −0.0000 | −0.0000 | −0.0000 |
| | (0.132) | (0.268) | (0.161) | (0.155) | (0.238) | (0.159) |
| Constant | −28.5864 *** | −29.1314 *** | −34.9458 *** | −29.7340 *** | −33.2832 *** | −34.4910 *** |
| | (0.000) | (0.000) | (0.000) | (0.000) | (0.000) | (0.000) |
| R-squared | 0.292 | 0.313 | 0.175 | 0.309 | 0.186 | 0.190 |
| **Panel B. State policy effects on changes in groceries and pharmacy mobility** | | | | | | |
| State policy | −0.4997 *** | −0.4636 *** | −0.2179 | −0.4077 *** | −0.0690 | −0.1761 |
| | (0.000) | (0.000) | (0.122) | (0.002) | (0.627) | (0.202) |
| Tests per population | −0.2149 * | −0.2237 * | −0.2636 * | −0.2396 * | −0.2723 * | −0.2988 ** |
| | (0.076) | (0.072) | (0.056) | (0.062) | (0.055) | (0.033) |
| State population | −0.0000 | −0.0000 | −0.0000 * | −0.0000 | −0.0000 | −0.0000 |
| | (0.102) | (0.284) | (0.087) | (0.154) | (0.215) | (0.156) |
| Constant | −1.0383 | −3.2244 | −9.6105 *** | −4.6980 * | −9.8218 *** | −10.0317 *** |
| | (0.715) | (0.223) | (0.000) | (0.072) | (0.002) | (0.000) |
| R-squared | 0.354 | 0.317 | 0.154 | 0.273 | 0.113 | 0.140 |
| **Panel C. State policy effects on changes in transit mobility** | | | | | | |
| State policy | −0.3284 *** | −0.4742 *** | −0.0439 | −0.3048 ** | 0.0610 | −0.1366 |
| | (0.008) | (0.000) | (0.739) | (0.014) | (0.642) | (0.285) |
| Tests per population | −0.3581 *** | −0.3436 *** | −0.3972 *** | −0.3708 *** | −0.4064 *** | −0.4156 *** |
| | (0.004) | (0.003) | (0.003) | (0.003) | (0.003) | (0.002) |
| State population | −0.0000 ** | −0.0000 ** | −0.0000 ** | −0.0000 ** | −0.0000 ** | −0.0000 ** |
| | (0.011) | (0.026) | (0.018) | (0.015) | (0.017) | (0.017) |
| Constant | −13.5521 ** | −10.4041 ** | −27.5296 *** | −17.6392 *** | −30.5015 *** | −26.4633 *** |
| | (0.038) | (0.050) | (0.000) | (0.002) | (0.000) | (0.000) |
| R-squared | 0.349 | 0.461 | 0.244 | 0.334 | 0.246 | 0.261 |
| **Panel D. State policy effects on changes in work mobility** | | | | | | |
| State policy | −0.3412 *** | −0.4815 *** | 0.0521 | −0.3644 *** | −0.1557 | −0.1428 |
| | (0.005) | (0.000) | (0.688) | (0.003) | (0.226) | (0.257) |
| Tests per population | −0.3788 *** | −0.3650 *** | −0.4264 *** | −0.3873 *** | −0.4072 *** | −0.4386 *** |
| | (0.002) | (0.001) | (0.001) | (0.001) | (0.002) | (0.001) |
| State population | −0.0000 *** | −0.0000 ** | −0.0000 ** | −0.0000 ** | −0.0000 ** | −0.0000 ** |
| | (0.009) | (0.023) | (0.024) | (0.011) | (0.033) | (0.015) |
| Constant | −34.2112 *** | −32.9007 *** | −41.5334 *** | −35.4065 *** | −38.4380 *** | −40.3752 *** |
| | (0.000) | (0.000) | (0.000) | (0.000) | (0.000) | (0.000) |
| R-squared | 0.376 | 0.486 | 0.264 | 0.393 | 0.284 | 0.281 |
| **Panel E. State policy effects on changes in overall mobility** | | | | | | |
| State policy | −0.4147 *** | −0.4147 *** | −0.4932 | −0.0472 *** | −0.4030 | −0.0964 |
| | (0.001) | (0.000) | (0.722) | (0.001) | (0.465) | (0.140) |
| Tests per population | −0.3304 *** | −0.3304 *** | −0.3247 *** | −0.3803 *** | −0.3446 *** | −0.3741 *** |
| | (0.006) | (0.004) | (0.005) | (0.004) | (0.005) | (0.002) |
| State population | −0.0000 *** | −0.0000 ** | −0.0000 ** | −0.0000 ** | −0.0000 ** | −0.0000 ** |
| | (0.008) | (0.027) | (0.019) | (0.012) | (0.032) | (0.016) |
| Constant | 4.0150 *** | 4.0170 *** | 1.6686 *** | 3.4157 *** | 2.0684 ** | 1.8886 *** |
| | (0.000) | (0.000) | (0.005) | (0.000) | (0.016) | (0.001) |
| R-squared | 0.399 | 0.466 | 0.232 | 0.391 | 0.239 | 0.265 |

The table presents the joint impact of state policy actions and COVID-19 testing on mobility trends in the US states. The dependent variables are the change in retail and recreation mobility, groceries and pharmacy mobility, transit mobility, work mobility, and overall mobility shown in Panels A-E, respectively. The *p*-values are reported beneath the coefficient estimates. *, **, and *** indicate significance at the 10%, 5%, and 1% levels, respectively.

### 5.3. State Policy Effects on Local Health Conditions

Social distancing to contain an infectious disease outbreak is viewed as an essential component of public health response. However, whether and to what extent various state policy measures are able to make a significant difference in subsequent disease levels is

an open empirical question. In this subsection, we test whether state restrictions improve subsequent health outcomes. To assess whether reduced mobility achieved through state mitigation interventions translates into beneficial health effects, we evaluated the effects of mitigation interventions on subsequent morbidity and mortality. We hypothesized that state policy actions would load with significantly negative coefficients in regressions where the dependent variables are subsequent morbidity and mortality rates. Table 4 presents results for the impact of state policy actions on subsequent COVID-19 cases and deaths in the US states using all 10 state policy measures. The dependent variable is the percentage change in cumulative cases and deaths from the end of the first quarter of 2020 to the end of the second quarter of 2020 shown in Panels A and B, respectively.

**Table 4.** State policy effects on local health conditions.

| | (1) | (2) | (3) | (4) | (5) | (6) |
|---|---|---|---|---|---|---|
| **State Policy** | State Restrictions | Stay Home | Traveler Quarantine | Business Closures | Gathering Bans | Postpone Elections |
| Panel A. State restriction effects on subsequent COVID cases | | | | | | |
| State policy | −0.5499 *** | −0.5167 *** | −0.1301 | −0.5063 *** | −0.2358 * | −0.1215 |
| | (0.000) | (0.001) | (0.357) | (0.000) | (0.098) | (0.391) |
| Hospital beds per population | 0.1247 | −0.6745 | 5.5619 | 1.8097 | 4.5476 | 6.1513 |
| | (0.971) | (0.857) | (0.138) | (0.595) | (0.223) | (0.103) |
| Constant | 56.2683 *** | 53.2092 *** | 12.7721 | 43.1994 *** | 25.8391 * | 10.7134 |
| | (0.000) | (0.001) | (0.228) | (0.001) | (0.063) | (0.294) |
| R-squared | 0.304 | 0.255 | 0.066 | 0.282 | 0.102 | 0.064 |
| Panel B. State restriction effects on subsequent COVID deaths | | | | | | |
| State policy | −0.1313 | −0.0497 | −0.0078 | −0.0855 | −0.2610 * | 0.1324 |
| | (0.404) | (0.763) | (0.957) | (0.574) | (0.073) | (0.362) |
| Hospital beds per population | −2.1661 | −0.6767 | 0.5564 | −0.7836 | −2.2757 | −0.1296 |
| | (0.795) | (0.939) | (0.943) | (0.923) | (0.766) | (0.987) |
| Constant | 71.6006 ** | 57.6883 | 49.6231 ** | 60.6039 ** | 84.5342 *** | 47.9845 ** |
| | (0.039) | (0.104) | (0.028) | (0.042) | (0.004) | (0.026) |
| R-squared | 0.015 | 0.002 | 0.000 | 0.007 | 0.065 | 0.017 |

The table presents the impact of state policy actions on subsequent COVID-19 cases and deaths in the US states. The dependent variable is the percentage change in cumulative cases and deaths from the end of the first quarter of 2020 to the end of the second quarter of 2020 in Panels A and B, respectively. The *p*-values are reported beneath the coefficient estimates. *, **, and *** indicate significance at the 10%, 5%, and 1% levels, respectively.

The results suggest some health gains associated with state policy actions in reducing coronavirus disease morbidity and mortality. The improvement in health conditions due to stricter state policies is generally restricted to the reduction in subsequent disease case incidences and with selected policy measures only, lending only partial support to our hypothesis above. Thus, the effect primarily works by alleviating the incidences of disease cases, not deaths from the disease. Moreover, not all state policy measures are equally deterrent. Among the social distancing measures, state restrictions, stay home orders, business closures, and gathering bans are the only effective mitigating interventions that help slow down the spread of the disease as measured by the incidences of COVID-19 cases. As for the incidences of COVID-19 deaths, the only state policy measure that is marginally significant is gathering bans. Traveler quarantines and primary election postponement policy measures, on the other hand, do not seem to make a statistically and economically significant difference in improving health outcomes.

### 5.4. Effects of State Policies and Disease Severity on Local Economic Conditions

What are the economic consequences of stricter state restrictions and disease severity? In this subsection, we test whether stricter state policies and/or higher disease severity result in deterioration in local economic conditions. The main goal of social distancing policies is to induce a slowdown in the level of economic activity to contain the virus. At least two important channels imply direct effects of health conditions on real economic outcomes. First, unhealthy individuals are less productive both when they are at work

or when they become sick and miss work. Second, bad health reduces human capital investment which is a major factor for economic growth both directly and indirectly through its adverse impact on life expectancy. We, therefore, hypothesized that state policies and higher disease severity would have significantly negative coefficient estimates in regressions where the dependent variables are measures of local economic conditions. Table 5 presents results on the joint effects of state policies as well as COVID-19 severity on real economic activity within the US states. The measure of economic conditions is the quarterly change in real GDP calculated from the last quarter of 2019 to the first quarter of 2020. COVID-19 case and death rates represent the average number of confirmed COVID-19 cases or deaths per 100,000 people in the state over the first quarter of 2020. Regressions control for the previous annualized quarterly change (from the third to fourth quarter of 2019) in real GDP. Panel A presents results on economic output using COVID-19 cases. Panel B presents the results using COVID-19 deaths.

**Table 5.** Effects of state policies and COVID-19 severity on local economic growth.

|  | (1) | (2) | (3) | (4) | (5) | (6) |
|---|---|---|---|---|---|---|
| **State Policy** | **State Restrictions** | **Stay Home** | **Traveler Quarantine** | **Business Closures** | **Gathering Bans** | **Postpone Elections** |
| Panel A. Effects of state policy and COVID-19 case severity on the change in real GDP | | | | | | |
| State policy | −0.4587 *** | −0.4987 *** | −0.0741 | −0.4729 *** | −0.1690 | −0.2013 |
|  | (0.000) | (0.000) | (0.585) | (0.000) | (0.209) | (0.174) |
| Case rate | −0.3667 *** | −0.3086 ** | −0.4539 *** | −0.3498 *** | −0.4058 *** | −0.3586 ** |
|  | (0.003) | (0.012) | (0.002) | (0.005) | (0.004) | (0.016) |
| Lagged change in Real GDP | −0.1303 | 0.0525 | −0.1995 | −0.1235 | −0.1345 | −0.2432 |
|  | (0.541) | (0.809) | (0.422) | (0.559) | (0.582) | (0.324) |
| Constant | −2.3063 *** | −2.7938 *** | −3.8812 *** | −2.6979 *** | −3.4986 *** | −3.8063 *** |
|  | (0.001) | (0.000) | (0.000) | (0.000) | (0.000) | (0.000) |
| R-squared | 0.390 | 0.407 | 0.190 | 0.401 | 0.212 | 0.216 |
| Panel B. Effects of state policy and COVID-19 death severity on the change in real GDP | | | | | | |
| State policy | −0.4598 *** | −0.4898 *** | −0.1055 | −0.4649 *** | −0.1458 | −0.1939 |
|  | (0.000) | (0.000) | (0.434) | (0.000) | (0.277) | (0.180) |
| Death rate | −0.4031 *** | −0.3380 *** | −0.4942 *** | −0.3758 *** | −0.4328 *** | −0.3936 *** |
|  | (0.001) | (0.006) | (0.001) | (0.002) | (0.002) | (0.007) |
| Lagged change in Real GDP | −0.1419 | 0.0399 | −0.2181 | −0.1318 | −0.1471 | −0.2509 |
|  | (0.496) | (0.852) | (0.372) | (0.527) | (0.543) | (0.302) |
| Constant | −2.3376 *** | −2.8519 *** | −3.8869 *** | −2.7706 *** | −3.6387 *** | −3.8554 *** |
|  | (0.001) | (0.000) | (0.000) | (0.000) | (0.000) | (0.000) |
| R-squared | 0.417 | 0.424 | 0.220 | 0.418 | 0.230 | 0.240 |

The table presents the impact of state policies and COVID-19 severity on real economic activity within the US states. Panels A and B show results using COVID-19 cases and deaths for the quarterly change in real GDP calculated from the last quarter of 2019 to the first quarter of 2020, respectively. Case rates and death rates represent the average number of confirmed COVID-19 cases or deaths per 100,000 people in the state over the first quarter of 2020. Regressions control for the previous quarterly change (from the third to fourth quarter of 2019) in real GDP. The *p*-values are reported beneath the coefficient estimates. *, **, and *** indicate significance at the 10%, 5%, and 1% levels, respectively.

What are the joint effects of stricter state restrictions and higher disease morbidity and mortality on the local economy? In both panels, focusing our attention on the social distancing measures, more restraining state restrictions result in larger contractions in real GDP with respect to the overall state restriction index as well as its stay home and business closure components. With regard to the real economic effects of the coronavirus disease, the results confirm the direct adverse effects of deteriorating health conditions on economic output. Higher incidences of coronavirus cases and deaths both lead to significant contraction of the state economy. The economic impact is slightly larger with incidences of deaths compared to incidences of cases. Thus, these results are generally consistent with our hypothesis that disease mitigation and disease severity measures lead to an economic slowdown in affected states.

The findings indicate that the direct impact of health conditions is relatively smaller than the effects of state policy measures. For example, in column 1 of both panels, state restrictions result in a 0.46 standard deviation drop in real economic output, whereas the corresponding decreases are 0.37 and 0.40 standard deviations with increasing COVID-19

cases and deaths, respectively. With respect to the direct impact of the disease on the economy, both cases and deaths are important determinants of the change in real GDP. Specifically, the standardized coefficient estimates are similar using both COVID-19 deaths and cases (the average across the columns indicates a decline by 0.40 standard deviations using COVID-19 deaths and a decline of 0.37 standard deviations using COVID-19 cases).

In additional robustness tests, we augmented our regression specifications with the interaction terms between (i) state policy and disease severity measures, and (ii) the quadratic terms of the lagged change in real GDP. We also ran panel regressions with state and quarter fixed effects for the initial four quarters of 2020 for which the state-level economic growth data are available. Our main results continue to prevail in all of these cases: social distancing policies and disease severity measures lead to an economic contraction in the affected states, alleviating the concerns that our results may be confounded by omitted variable bias or specification problems in our empirical methodology.

*5.5. Predictive Ability of State Policies and COVID-19 Severity for Forecasting Local Economic Contractions*

The goal of this subsection is to quantify the relative predictive power of various state policies and disease severity measures in forecasting local economic contractions. We hypothesized that the adverse effects of state policies and disease severity on local economies are not homogenous. In order to evaluate the predictive utility of our target variables in forecasting local economic contraction, we utilized a machine learning approach. We chose machine learning because prediction and classification are the domains in which machine learning algorithms excel. Machine learning models are performance-driven with a focus on the predictive power as well as classification accuracy and stability, based on known properties learned from the training samples. Table 6 presents the performance metrics of the classifier performance obtained for each model. Figure 1 illustrates the feature importance rankings derived from the classifier's coefficients.

In Table 6, we observe a great deal of heterogeneity in the predictive power of disease mitigation and disease severity measures, consistent with our hypothesis. The highest explanatory power in predicting real economic conditions arises from state restrictions, stay home orders, business closures, disease cases, and deaths. Figure 1 further shows that when a horse-race is run between state policy measures and disease severity, included in the models separately, both disease cases and deaths perform significantly better than social distancing measures in the majority of the specifications. The only exception is the primary election postponement, whose predictive ability outweighs both disease cases and deaths. When disease cases and deaths are jointly included in the model, the predictive ability of state restrictions and business closures measures strictly dominates the forecasting power of disease severity, but primary election postponement loses some of its predictive ability to deaths from the disease, which now has more predictive power. Overall, our conclusion that disease severity outperforms social distancing measures in predicting real economic conditions holds for 14 out of 18 specifications. When the relative predictive abilities of social distancing measures are evaluated on their own, gathering bans and traveler quarantines perform the worst, followed by stay home orders. Business closures and primary election postponement have the highest explanatory power in forecasting real economic contraction outcomes. Controlling for disease severity does not affect our general conclusion that predictions with business closures and primary election postponement are superior relative to the remaining social distancing measures. Importantly, when controlled for separately, both disease cases and deaths are the most important determinants of real economic output contraction occurrence. When both cases and deaths are accounted for, deaths from the disease perform the best, followed by business closures, primary election postponement, and disease cases.

Collectively, these results suggest that the direct negative impact of the disease itself is a key factor in forecasting real economic contraction occurrence during the current pandemic. Social distancing measures are also important, but their effects are not uniform. While state restrictions, business closures, and primary election postponement are first-

order determinants of the contraction outcomes in the real economic output, stay home orders, gathering bans, and traveler quarantines have only trivial predictive capability.

**Table 6.** Predictive modeling of state policies and COVID-19 severity for forecasting real economic contraction in local economies.

| Target Features | Model Accuracy | Baseline Accuracy | Model $F_1$ | Baseline $F_1$ |
|---|---|---|---|---|
| State restrictions | 0.627 | 0.431 | 0.568 | 0.601 |
| Stay home | 0.607 | 0.431 | 0.664 | 0.601 |
| Traveler quarantine | 0.564 | 0.431 | 0.486 | 0.601 |
| Business closures | 0.571 | 0.431 | 0.653 | 0.601 |
| Gathering bans | 0.627 | 0.431 | 0.579 | 0.601 |
| Postpone elections | 0.727 | 0.431 | 0.592 | 0.601 |
| COVID cases | 0.685 | 0.431 | 0.567 | 0.601 |
| COVID deaths | 0.607 | 0.431 | 0.381 | 0.601 |
| State restrictions, COVID cases | 0.745 | 0.431 | 0.681 | 0.601 |
| Stay home, COVID cases | 0.645 | 0.431 | 0.554 | 0.601 |
| Traveler quarantine, COVID cases | 0.687 | 0.431 | 0.587 | 0.601 |
| Business closures, COVID cases | 0.629 | 0.431 | 0.596 | 0.601 |
| Gathering bans, COVID cases | 0.665 | 0.431 | 0.575 | 0.601 |
| Postpone elections, COVID cases | 0.667 | 0.431 | 0.534 | 0.601 |
| State restrictions, COVID deaths | 0.745 | 0.431 | 0.675 | 0.601 |
| Stay home, COVID deaths | 0.607 | 0.431 | 0.445 | 0.601 |
| Traveler quarantine, COVID deaths | 0.645 | 0.431 | 0.514 | 0.601 |
| Business closures, COVID deaths | 0.551 | 0.431 | 0.510 | 0.601 |
| Gathering bans, COVID deaths | 0.627 | 0.431 | 0.483 | 0.601 |
| Postpone elections, COVID deaths | 0.707 | 0.431 | 0.574 | 0.601 |
| State restrictions, COVID cases, COVID deaths | 0.705 | 0.431 | 0.631 | 0.601 |
| Stay home, COVID cases, COVID deaths | 0.607 | 0.431 | 0.477 | 0.601 |
| Traveler quarantine, COVID cases, COVID deaths | 0.665 | 0.431 | 0.575 | 0.601 |
| Business closures, COVID cases, COVID deaths | 0.629 | 0.431 | 0.596 | 0.601 |
| Gathering bans, COVID cases, COVID deaths | 0.665 | 0.431 | 0.518 | 0.601 |
| Postpone elections, COVID cases, COVID deaths | 0.667 | 0.431 | 0.534 | 0.601 |
| Stay home, Traveler quarantine, Business closures, Gathering bans, Postpone elections | 0.569 | 0.431 | 0.490 | 0.601 |
| Stay home, Traveler quarantine, Business closures, Gathering bans, Postpone elections, COVID cases | 0.625 | 0.431 | 0.510 | 0.601 |
| Stay home, Traveler quarantine, Business closures, Gathering bans, Postpone elections, COVID deaths | 0.587 | 0.431 | 0.444 | 0.601 |
| Stay home, Traveler quarantine, Business closures, Gathering bans, Postpone elections, COVID cases, COVID deaths | 0.605 | 0.431 | 0.477 | 0.601 |

The table presents performance metrics for the predictive modeling using machine learning and the relative importance of social policies and COVID-19 severity in forecasting real economic contraction in local economies. All models control for the previous quarterly change (from the third to fourth quarter of 2019) in real GDP. The table presents accuracy and $F_1$ metrics for a variety of combinations of the selected target features. Baseline accuracy and $F_1$ metrics are presented next to the actual classifier performance for each model.

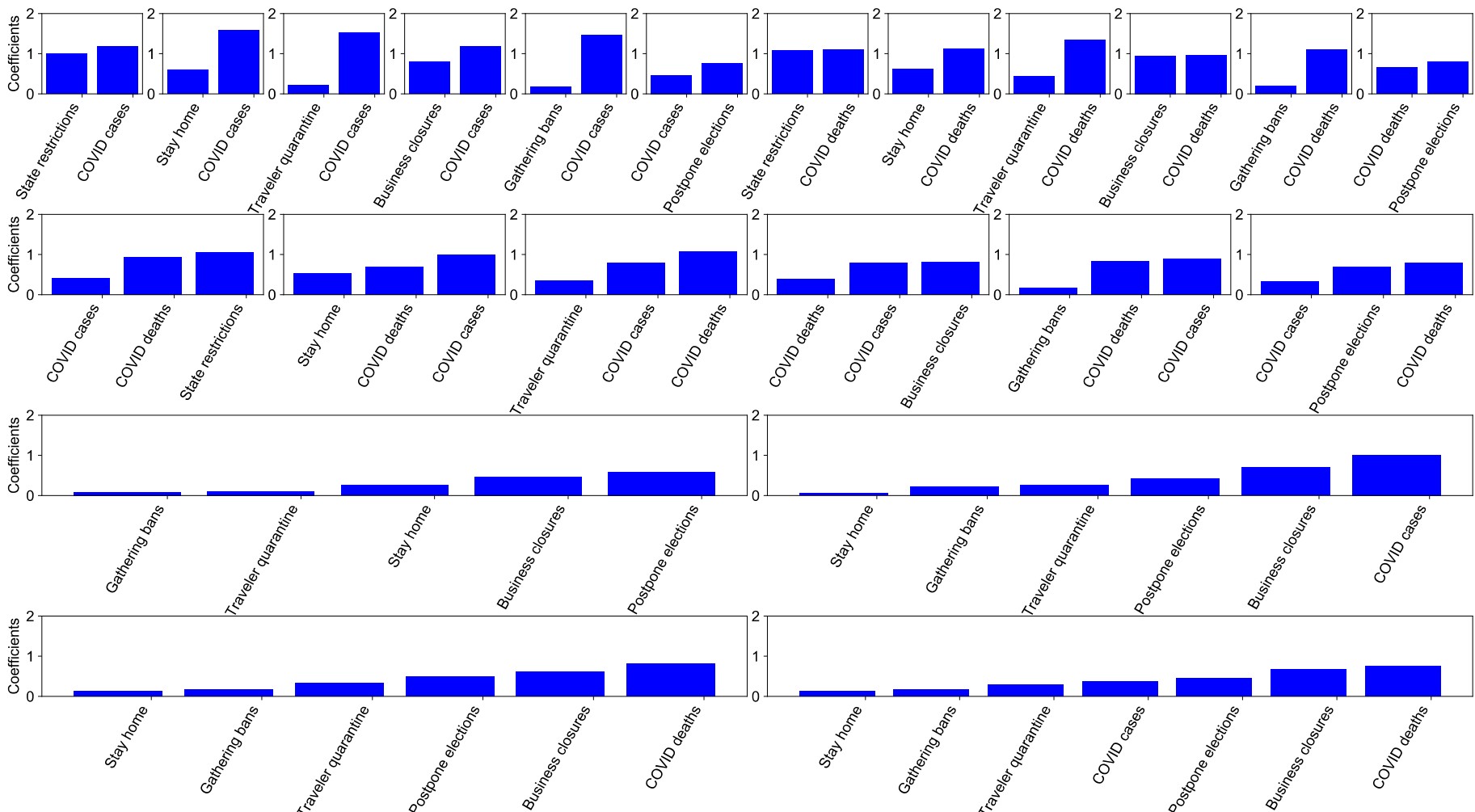

**Figure 1.** Feature importance rankings from machine learning evaluated models. The figure presents the relative importance rankings of machine learning for each model containing more than one feature from Table 6.

### 5.6. Effects of State Policies and Disease Severity on Local Economic Activity

Having shown that both state policy actions and disease severity are important in explaining the cross-sectional differences in changes in real economic conditions, next, we evaluated specific channels of impact. We argue that state policy measures and disease severity can influence local economic conditions through different channels. One potential economic mechanism is the labor market. We hypothesized that the adverse effects of state policies and disease severity on local economies through the labor market channel are not homogenous. To test this conjecture, in Table 7, we assessed the impact of state policies and disease severity on local economic activity using the state coincident index. The state coincident index is based on labor market data and combines four indicators of economic conditions: nonfarm payroll employment, average hours worked in manufacturing by production workers, the unemployment rate, and wage and salary disbursements plus proprietors' income. Panel A shows results using COVID-19 cases. Panel B shows results using COVID-19 deaths.

**Table 7.** Effects of state policies and COVID-19 severity on local economic activity.

| State Policy | (1) State Restrictions | (2) Stay Home | (3) Traveler Quarantine | (4) Business Closures | (5) Gathering Bans | (6) Postpone Elections |
|---|---|---|---|---|---|---|
| Panel A. Effects of state policy and COVID-19 case severity on the change in the state coincident index | | | | | | |
| State policy | −0.4593 *** | −0.3005 ** | −0.3401 ** | −0.2334 | 0.0675 | −0.1839 |
| | (0.001) | (0.038) | (0.018) | (0.110) | (0.646) | (0.275) |
| Case rate | 0.1022 | 0.1016 | −0.0174 | 0.0795 | 0.0328 | 0.1206 |
| | (0.435) | (0.473) | (0.900) | (0.578) | (0.823) | (0.451) |
| Lagged change in state coincident index | 0.4491 | 0.7322 | 0.6887 | 0.7306 | 0.8773 | 0.6600 |
| | (0.373) | (0.168) | (0.189) | (0.178) | (0.115) | (0.249) |
| Constant | 0.0117 | −0.0004 | −0.0078 | −0.0046 | −0.0180 ** | −0.0125 ** |
| | (0.185) | (0.962) | (0.134) | (0.551) | (0.043) | (0.015) |
| R-squared | 0.246 | 0.137 | 0.162 | 0.103 | 0.055 | 0.075 |
| Panel B. Effects of state policy and COVID-19 death severity on the change in the state coincident index | | | | | | |
| State policy | −0.4577 *** | −0.3078 ** | −0.3378 ** | −0.2399 * | 0.0599 | −0.1911 |
| | (0.001) | (0.033) | (0.020) | (0.099) | (0.685) | (0.245) |
| Death rate | 0.1222 | 0.1387 | −0.0036 | 0.1169 | 0.0665 | 0.1513 |
| | (0.351) | (0.329) | (0.980) | (0.416) | (0.654) | (0.337) |
| Lagged change in state coincident index | 0.4816 | 0.7694 | 0.6941 | 0.7634 | 0.8995 | 0.6920 |
| | (0.341) | (0.148) | (0.189) | (0.161) | (0.108) | (0.225) |
| Constant | 0.0115 | −0.0003 | −0.0081 | −0.0046 | −0.0180 ** | −0.0126 ** |
| | (0.190) | (0.968) | (0.118) | (0.541) | (0.041) | (0.013) |
| R-squared | 0.250 | 0.145 | 0.161 | 0.110 | 0.058 | 0.083 |

The table presents the impact of state policies and COVID-19 severity on local economic activity within the US states using the state coincident index. Panel A shows results using state policy measures and COVID-19 cases. Panel B shows results using state policy measures and COVID-19 deaths. The *p*-values are reported beneath the coefficient estimates. *, **, and *** indicate significance at the 10%, 5%, and 1% levels, respectively.

In Table 7, we observe a great deal of heterogeneity in the importance of state policies and disease severity measures for the deterioration in the economic conditions as measured by the state coincident index, consistent with our hypothesis. Our primary finding from this investigation is that neither disease cases nor deaths matter in explaining the changes in state economic conditions as measured by the state coincident index. Three out of six social distancing measures are significant in both panels: state restrictions, stay home orders, and traveler quarantine. Albeit weaker, there is also some support for the relevance of business closures in Panel B. The combined measure of state restrictions has the largest economic impact (0.46 standard deviations), followed by traveler quarantines (0.34 standard deviation), stay home orders (0.30–0.31 standard deviations), and business closures

(0.24 standard deviation). Thus, states that have adopted more strict social distancing measures have experienced more severe contractions in their coincident index. Health conditions do not exert a significant impact on state economies.

### 5.7. Effects of State Policies and Disease Severity on Local Business Earnings and Employment

Poor health conditions may distress an individual and social welfare by reducing earnings capacity and hours worked (Bartel and Taubman 1979). Considering the insignificance of health conditions in explaining the cross-sectional differences in the state coincident index, a more refined empirical assessment as to which component(s) of state economic outcomes are affected more by disease severity is warranted: labor productivity as measured by either number of hours worked or wage and salary disbursements as opposed to the level of state employment. Our next goal was to evaluate the relative importance of these two competing forces. We hypothesized that the adverse effects of state policies on local economies operate through both labor productivity and state employment. On the other hand, since disease severity did not have a statistically significant impact on the state coincident index (see Section 5.6), we did not expect any of these two forces to be in effect when disease severity is considered. Tables 8 and 9 present the effects of state policies and disease severity on local business earnings and employment for low-income earning workers, respectively. In both tables, Panels A and C show results using COVID-19 cases for all businesses and small businesses, respectively. Panels B and D show results using COVID-19 deaths for businesses and small businesses, respectively.

In both tables, when all firms are considered in Panels A and B, state restrictions, stay home, business closures, and primary election postponement matter with an economic magnitude ranging from 0.27–0.37. When small businesses are considered in Panels C and D, with the exception of primary election postponement, the same social distancing measures continue to be significant determinants of both business earnings and employment. The largest effect comes from the stay home orders (0.31–0.39 standard deviations) and a slightly smaller impact is observed using state restrictions (0.26–0.29 standard deviations) and business closures (about 0.29–0.31 standard deviations). Thus, consistent with the results on the state coincident index, state policy measures are major determinants of disruptions in the economic activity level within the US states. On the other hand, disease severity continues to exert no direct impact on state economic outcomes as measured by either the level of business earnings or the level of employment. These findings lend support to our hypothesis above: while the adverse effects of state policies on local economies operate through both labor productivity and state employment channels, neither of these economic forces are relevant for the impact of disease severity on local economic conditions.

### 5.8. Effects of State Policies and Disease Severity on Local Unemployment and Bankruptcy Filings

The COVID-19 shock has caused a sudden increase in fear, anxiety, and uncertainty across US states. Higher disease morbidity and mortality lead to a spike in risk aversion. This heightened health risk would increase employees' risk of unemployment. During disease outbreaks, distressed firms are forced to discharge workers to cover cash shortfalls (Ofek 1993). The firing costs incurred as a result of discharging workers would also increase the costs of financial distress. We hypothesized that the adverse effects of disease severity on local economies operate through state unemployment and bankruptcy filings. On the other hand, considering that labor productivity and employment channels are already in effect for state mitigation measures, we did not expect a significant impact for state policy measures through the unemployment and bankruptcy channels. We evaluated the impact of state policies and disease severity on state unemployment and bankruptcy filings and present our results in Table 10. Panel A shows effects on state unemployment using state policy measures and COVID-19 cases. Panel B shows effects on state unemployment using state policy measures and COVID-19 deaths. Panel C shows effects on bankruptcy filings using state policy measures and COVID-19 cases. Panel D shows effects on bankruptcy filings using state policy measures and COVID-19 deaths.

**Table 8.** Effects of state policies and COVID-19 severity on local business earnings.

| | (1) | (2) | (3) | (4) | (5) | (6) |
|---|---|---|---|---|---|---|
| **State Policy** | State Restrictions | Stay Home | Traveler Quarantine | Business Closures | Gathering Bans | Postpone Elections |
| Panel A. Effects of state policy and COVID-19 case severity on the change in business earnings | | | | | | |
| State policy | −0.2817 ** | −0.3701 *** | 0.0560 | −0.2929 ** | −0.0807 | −0.2831 * |
| | (0.048) | (0.010) | (0.700) | (0.038) | (0.577) | (0.053) |
| Case rate | 0.0142 | 0.0106 | −0.0314 | −0.0060 | −0.0254 | −0.0014 |
| | (0.920) | (0.938) | (0.830) | (0.966) | (0.861) | (0.992) |
| Lagged change in Real GDP | −0.0085 | −0.0052 | −0.0089 | −0.0086 | −0.0087 | −0.0125 * |
| | (0.219) | (0.445) | (0.216) | (0.209) | (0.225) | (0.081) |
| Constant | 0.0310 | 0.0300 | 0.0007 | 0.0263 | 0.0101 | 0.0149 |
| | (0.150) | (0.112) | (0.966) | (0.180) | (0.637) | (0.388) |
| R-squared | 0.112 | 0.164 | 0.037 | 0.120 | 0.041 | 0.109 |
| Panel B. Effects of state policy and COVID-19 death severity on the change in business earnings | | | | | | |
| State policy | −0.2718 * | −0.3660 *** | 0.0537 | −0.2857 ** | −0.0675 | −0.2892 ** |
| | (0.055) | (0.010) | (0.710) | (0.049) | (0.643) | (0.047) |
| Death rate | −0.0729 | −0.0906 | −0.1031 | −0.0297 | −0.0931 | −0.1166 |
| | (0.601) | (0.500) | (0.474) | (0.835) | (0.521) | (0.400) |
| Lagged change in Real GDP | −0.0089 | −0.0057 | −0.0091 | −0.0087 | −0.0090 | −0.0130 * |
| | (0.194) | (0.400) | (0.203) | (0.202) | (0.209) | (0.067) |
| Constant | 0.0342 | 0.0345 * | 0.0041 | 0.0268 | 0.0120 | 0.0207 |
| | (0.111) | (0.067) | (0.811) | (0.156) | (0.559) | (0.231) |
| R-squared | 0.117 | 0.172 | 0.047 | 0.120 | 0.049 | 0.122 |
| Panel C. Effects of state policy and COVID-19 case severity on the change in small business earnings | | | | | | |
| State policy | −0.2914 ** | −0.3853 *** | 0.0257 | −0.3059 ** | −0.0912 | −0.2233 |
| | (0.043) | (0.007) | (0.861) | (0.032) | (0.532) | (0.134) |
| Case rate | 0.0050 | 0.0016 | −0.0387 | −0.0157 | −0.0359 | −0.0171 |
| | (0.972) | (0.991) | (0.793) | (0.911) | (0.806) | (0.906) |
| Lagged change in Real GDP | −0.0053 | −0.0020 | −0.0058 | −0.0054 | −0.0055 | −0.0085 |
| | (0.422) | (0.754) | (0.397) | (0.408) | (0.425) | (0.223) |
| Constant | 0.0254 | 0.0245 | −0.0034 | 0.0210 | 0.0054 | 0.0066 |
| | (0.218) | (0.172) | (0.838) | (0.264) | (0.792) | (0.694) |
| R-squared | 0.099 | 0.157 | 0.017 | 0.109 | 0.024 | 0.063 |
| Panel D. Effects of state policy and COVID-19 death severity on the change in small business earnings | | | | | | |
| State policy | −0.2805 ** | −0.3813 *** | 0.0228 | −0.2945 ** | −0.0751 | −0.2317 |
| | (0.049) | (0.007) | (0.875) | (0.044) | (0.608) | (0.116) |
| Death rate | −0.0933 | −0.1115 | −0.1242 | −0.0488 | −0.1135 | −0.1352 |
| | (0.505) | (0.408) | (0.393) | (0.734) | (0.438) | (0.341) |
| Lagged change in Real GDP | −0.0057 | −0.0025 | −0.0060 | −0.0055 | −0.0057 | −0.0089 |
| | (0.384) | (0.699) | (0.377) | (0.398) | (0.401) | (0.194) |
| Constant | 0.0288 | 0.0293 | 0.0004 | 0.0215 | 0.0074 | 0.0123 |
| | (0.161) | (0.101) | (0.980) | (0.234) | (0.707) | (0.463) |
| R-squared | 0.108 | 0.169 | 0.031 | 0.111 | 0.036 | 0.080 |

The table presents the impact of state policies and COVID-19 severity on business earnings within the US states. Panels A and C show results using state policy measures and COVID-19 cases for all businesses and small businesses, respectively. Panels B and D show results using state policy measures and COVID-19 deaths for businesses and small businesses, respectively. The *p*-values are reported beneath the coefficient estimates. *, **, and *** indicate significance at the 10%, 5%, and 1% levels, respectively.

**Table 9.** Effects of state policies and COVID-19 severity on local business employment.

| State Policy | (1) State Restrictions | (2) Stay Home | (3) Traveler Quarantine | (4) Business Closures | (5) Gathering Bans | (6) Postpone Elections |
|---|---|---|---|---|---|---|
| Panel A. Effects of state policy and COVID-19 case severity on the change in business employment | | | | | | |
| State policy | −0.2964 ** | −0.3368 ** | 0.0125 | −0.3049 ** | −0.0261 | −0.2811 * |
| | (0.037) | (0.019) | (0.931) | (0.030) | (0.857) | (0.055) |
| Case rate | 0.0091 | 0.0002 | −0.0339 | −0.0124 | −0.0325 | −0.0088 |
| | (0.948) | (0.999) | (0.817) | (0.929) | (0.823) | (0.950) |
| Lagged change in Real GDP | −0.0067 | −0.0046 | −0.0071 | −0.0068 | −0.0071 | −0.0096 * |
| | (0.182) | (0.364) | (0.174) | (0.173) | (0.178) | (0.066) |
| Constant | 0.0212 | 0.0175 | −0.0009 | 0.0174 | 0.0012 | 0.0083 |
| | (0.173) | (0.203) | (0.944) | (0.221) | (0.939) | (0.508) |
| R-squared | 0.126 | 0.147 | 0.040 | 0.132 | 0.040 | 0.113 |
| Panel B. Effects of state policy and COVID-19 death severity on the change in business employment | | | | | | |
| State policy | −0.2808 ** | −0.3316 ** | 0.0105 | −0.2828 ** | −0.0033 | −0.2909 ** |
| | (0.044) | (0.018) | (0.941) | (0.049) | (0.982) | (0.043) |
| Death rate | −0.1315 | −0.1513 | −0.1623 | −0.0900 | −0.1618 | −0.1763 |
| | (0.339) | (0.262) | (0.257) | (0.524) | (0.263) | (0.200) |
| Lagged change in Real GDP | −0.0071 | −0.0050 | −0.0074 | −0.0070 | −0.0074 | −0.0101 ** |
| | (0.150) | (0.308) | (0.151) | (0.157) | (0.151) | (0.049) |
| Constant | 0.0249 | 0.0225 * | 0.0035 | 0.0187 | 0.0040 | 0.0144 |
| | (0.107) | (0.100) | (0.776) | (0.170) | (0.785) | (0.246) |
| R-squared | 0.143 | 0.170 | 0.065 | 0.140 | 0.065 | 0.144 |
| Panel C. Effects of state policy and COVID-19 case severity on the change in small business employment | | | | | | |
| State policy | −0.2761 ** | −0.3126 ** | 0.0380 | −0.3022 ** | −0.1305 | −0.2161 |
| | (0.050) | (0.028) | (0.792) | (0.030) | (0.360) | (0.139) |
| Case rate | 0.0568 | 0.0485 | 0.0139 | 0.0380 | 0.0181 | 0.0363 |
| | (0.684) | (0.724) | (0.923) | (0.782) | (0.899) | (0.797) |
| Lagged change in Real GDP | −0.0076 | −0.0057 | −0.0079 | −0.0076 | −0.0075 | −0.0098 * |
| | (0.110) | (0.230) | (0.110) | (0.103) | (0.124) | (0.050) |
| Constant | 0.0200 | 0.0167 | −0.0001 | 0.0177 | 0.0091 | 0.0073 |
| | (0.173) | (0.199) | (0.995) | (0.185) | (0.529) | (0.543) |
| R-squared | 0.133 | 0.151 | 0.059 | 0.149 | 0.075 | 0.101 |
| Panel D. Effects of state policy and COVID-19 death severity on the change in small business employment | | | | | | |
| State policy | −0.2606 * | −0.3048 ** | 0.0404 | −0.2937 ** | −0.1189 | −0.2185 |
| | (0.062) | (0.031) | (0.776) | (0.040) | (0.407) | (0.131) |
| Death rate | −0.0702 | −0.0887 | −0.0991 | −0.0238 | −0.0820 | −0.1093 |
| | (0.610) | (0.512) | (0.486) | (0.866) | (0.565) | (0.432) |
| Lagged change in Real GDP | −0.0080 * | −0.0062 | −0.0082 * | −0.0079 * | −0.0079 | −0.0103 ** |
| | (0.087) | (0.185) | (0.093) | (0.090) | (0.105) | (0.038) |
| Constant | 0.0232 | 0.0210 | 0.0036 | 0.0194 | 0.0117 | 0.0123 |
| | (0.113) | (0.108) | (0.757) | (0.132) | (0.399) | (0.306) |

The table presents the impact of state policies and COVID-19 severity on business employment within the US states. Panels A and C show results using state policy measures and COVID-19 cases for all businesses and small businesses, respectively. Panels B and D show results using state policy measures and COVID-19 deaths for businesses and small businesses, respectively. The *p*-values are reported beneath the coefficient estimates. *, **, and *** indicate significance at the 10%, 5%, and 1% levels, respectively.

While mandatory social distancing restrictions do not seem to adversely affect state unemployment, disease severity in terms of both cases and deaths has a significant impact. In Panel A, the economic impact of COVID-19 cases ranges from 0.26 to 0.32. Similarly, in Panel B, the negative effects of COVID-19 deaths on local unemployment range from 0.26 to 0.29. The immediate increase in the official unemployment rate is greatly understated to the extent that individuals who would like to work do not report themselves as unemployed while waiting for the social distancing measures to subside and job search becomes feasible again. Since not all individuals who lose their jobs are eligible to receive unemployment insurance benefits or are able to quickly receive them, new unemployment insurance claims should be viewed as a lower-bound estimate of initial job losses (Petrosky-Nadeau and Valletta 2020). In Panels C and D, qualitatively similar and quantitatively stronger results hold for total bankruptcy filings. With the exception of primary election postponement, which is marginally significant in Panel C, mandatory social distancing measures imposed

by the states do not seem to adversely affect personal and business bankruptcies. On the other hand, COVID-19 cases and deaths have significant negative influences on the total number of bankruptcies in all specifications, with an economic magnitude ranging from 0.42 to 0.56 for COVID-19 cases and 0.35-0.48 for COVID-19 deaths. Overall, consistent with our hypothesis, our results confirm important negative supply-effects of disease severity by spiking unemployment and the total number of bankruptcies, whereas state policies do not seem to affect local economies through the unemployment and bankruptcies channel.

**Table 10.** Effects of state policies and COVID-19 severity on local unemployment and bankruptcy filings.

| | (1) | (2) | (3) | (4) | (5) | (6) |
|---|---|---|---|---|---|---|
| **State Policy** | **State Restrictions** | **Stay Home** | **Traveler Quarantine** | **Business Closures** | **Gathering Bans** | **Postpone Elections** |
| Panel A. Effects of state policy and COVID-19 case severity on the change in state unemployment | | | | | | |
| State policy | −0.1399 | −0.0468 | −0.1880 | −0.1637 | 0.0330 | 0.0908 |
| | (0.316) | (0.741) | (0.178) | (0.242) | (0.814) | (0.555) |
| Case rate | 0.3127 ** | 0.3022 ** | 0.2593 * | 0.3210 ** | 0.2870 ** | 0.2532 |
| | (0.028) | (0.037) | (0.066) | (0.024) | (0.046) | (0.103) |
| Constant | 3.6389 *** | 3.0607 *** | 3.1814 *** | 3.5567 *** | 2.6484 *** | 2.7718 *** |
| | (0.000) | (0.000) | (0.000) | (0.000) | (0.003) | (0.000) |
| R-squared | 0.105 | 0.088 | 0.120 | 0.111 | 0.087 | 0.092 |
| Panel B. Effects of state policy and COVID-19 death severity on the change in state unemployment | | | | | | |
| State policy | −0.1313 | −0.0428 | −0.1828 | −0.1624 | 0.0259 | 0.1071 |
| | (0.349) | (0.764) | (0.198) | (0.250) | (0.856) | (0.485) |
| Death rate | 0.2856 ** | 0.2779 * | 0.2279 | 0.2985 ** | 0.2634 * | 0.2242 |
| | (0.045) | (0.056) | (0.111) | (0.037) | (0.070) | (0.147) |
| Constant | 3.7335 *** | 3.1786 *** | 3.3058 *** | 3.6943 *** | 2.8213 *** | 2.8784 *** |
| | (0.000) | (0.000) | (0.000) | (0.000) | (0.001) | (0.000) |
| R-squared | 0.089 | 0.074 | 0.104 | 0.098 | 0.073 | 0.082 |
| Panel C. Effects of state policy and COVID-19 case severity on the change in bankruptcy filings | | | | | | |
| State policy | −0.0012 | −0.0405 | 0.0223 | −0.0016 | −0.1880 | 0.2553 * |
| | (0.993) | (0.764) | (0.865) | (0.990) | (0.147) | (0.072) |
| Case rate | −0.4607 *** | −0.4503 *** | −0.4563 *** | −0.4606 *** | −0.4244 *** | −0.5624 *** |
| | (0.001) | (0.002) | (0.001) | (0.001) | (0.002) | (0.000) |
| Lagged change in Real GDP | 0.0080 | 0.0088 | 0.0082 | 0.0080 | 0.0102 | 0.0115 |
| | (0.449) | (0.418) | (0.439) | (0.449) | (0.328) | (0.267) |
| Constant | −0.0497 | −0.0455 | −0.0517 * | −0.0497 * | −0.0243 | −0.0618 ** |
| | (0.127) | (0.111) | (0.054) | (0.091) | (0.407) | (0.013) |
| R-squared | 0.241 | 0.242 | 0.241 | 0.241 | 0.274 | 0.292 |
| Panel D. Effects of state policy and COVID-19 death severity on the change in bankruptcy filings | | | | | | |
| State policy | −0.0174 | −0.0565 | 0.0160 | −0.0096 | −0.1863 | 0.2133 |
| | (0.896) | (0.686) | (0.907) | (0.943) | (0.168) | (0.145) |
| Death rate | −0.3964 *** | −0.3839 *** | −0.3947 *** | −0.3969 *** | −0.3546 ** | −0.4799 *** |
| | (0.005) | (0.008) | (0.006) | (0.006) | (0.012) | (0.001) |
| Lagged change in Real GDP | 0.0091 | 0.0102 | 0.0093 | 0.0091 | 0.0114 | 0.0121 |
| | (0.401) | (0.363) | (0.400) | (0.403) | (0.292) | (0.265) |
| Constant | −0.0564 * | −0.0531 * | −0.0607 ** | −0.0582 * | −0.0342 | −0.0705 *** |
| | (0.092) | (0.070) | (0.027) | (0.053) | (0.256) | (0.006) |
| R-squared | 0.190 | 0.192 | 0.190 | 0.190 | 0.222 | 0.226 |

The table presents the impact of state policies and COVID-19 severity on local unemployment and bankruptcies within the US states. Panel A shows results on state unemployment using state policy measures and COVID cases. Panel B shows results on state unemployment using state policy measures and COVID deaths. Panel C shows results on bankruptcy filings using state policy measures and COVID-19 cases. Panel D shows results on bankruptcy filings using state policy measures and COVID-19 deaths. The *p*-values are reported beneath the coefficient estimates. *, **, and *** indicate significance at the 10%, 5%, and 1% levels, respectively.

## 6. Conclusions

The current pandemic continues to have dramatic effects on the world population with strong repercussions on the world economy. The debate on the desirability of an economic lockdown to mitigate the COVID-19 pandemic is an important one. While public health experts downplay the tradeoff between health and economic outcomes, academics and politicians draw attention to the possible dichotomy between the measures to improve

health outcomes during the pandemic and the resulting economic burden. Do policy actions to contain the spread of the COVID-19 disease save lives? Do such actions have detrimental effects on the economic conditions of an average citizen? Our current investigation is among the first to provide empirical evidence on these important questions, and the answers appear to show important tradeoffs between economic and health outcomes. Our findings indicate that state policy measures are effective in decreasing mobility and slowing down the spread of the virus but at the cost of significant and immediate economic contractions. The economic impact of the disease operates through unemployment and bankruptcy channels, whereas lockdown measures cause immediate contractions in local economic activity.

Surely, there is an inevitable trade-off between the severity of the short-run economic contraction caused by the epidemic and the health consequences of that epidemic. Dealing with this trade-off is a key challenge confronting policymakers. The threat of disease spread that induces people to voluntarily and involuntarily cut back on consumption and labor reduces the severity of the epidemic, as measured by the number of cases and deaths, but these decisions also exacerbate the size of the economic contraction caused by the epidemic. The best containment policies would not only save lives but should also decrease the severity of the economic recession. This article contributes to the research and policy debates on the costs and benefits of lockdown measures to mitigate the contagion of the disease.

Pandemics, like many other natural disasters, offer a unique opportunity to causally evaluate the economic impact of health risks by providing a randomized control trial at a large scale and helping to isolate exogenous variations in health conditions. In this paper, we particularly focus on the spatial and behavioral management of the COVID-19 health crisis by state governments. These measures are intended to minimize the disease contagion by social distancing and flattening the curve to avoid the breakdown in the services of medical facilities. Examples of such measures are restaurant and bar closures, cancellation of social events, shelter-in-place orders, remote work arrangements, or business closures. Most spatial restrictions are initiated by state and local government authorities and their efficiency in reducing mobility and thereby ameliorating the severity of the disease and the speed of its transmission is thus not certain.

In an effort to aid evidence-based policy responses, in this paper, we assess the health and economic outcomes of state social distancing measures. We document the cross-sectional patterns of social distancing measures across states. We further show how states differ in their average impact of an outbreak and how sensitive their local economy is to the pandemic disease. In summary, we provide novel insight into the extent to which the pandemic shock differently affects the policy makers' reaction to the threat of the virus contagion and the economic and health burden associated with such actions. Initial economic responses within the individual US states indicate that business closures, primary election postponement, and state restrictions and social distancing measures were the most detrimental to the real local economies in the short term.

Our findings suggest that states with higher death incidences have imposed stricter restrictions broadly and more restrictive stay home orders, business closures, and primary election postponement more specifically. While deaths are an important determinant of the severity of state actions, with the exception of election postponement, cases alone do not affect the stringency of the state mitigation measures. Except for traveler quarantines, gathering bans, and primary election postponement, whose impact on mobility is insignificant, the primary channel of impact for the health and economic consequences of state policy actions is to restrict individual behavior and movement. Among social distancing measures, state restrictions, stay home orders, business closures, and gathering bans are the most effective social interventions that help slow down disease cases. As for the incidences of disease deaths, the only state policy measure that is useful is gathering bans.

In conclusion, health conditions and lockdown measures are both important for the real economy but through strikingly different channels. The impact of disease severity on

the real economy comes primarily through state unemployment and personal and business bankruptcies. Social distancing measures, on the other hand, are significant determinants of the general economic activity level as well as business earnings and employment within the US states. With respect to the relative severity of the disease and social policy outcomes, deaths from the disease have a bigger negative impact compared to disease cases, but the impact of health conditions is relatively smaller than the effects of state policy measures. Disease severity, on the other hand, can better predict real local economic contraction outcomes relative to mandatory social distancing policy measures.

**Author Contributions:** Conceptualization, I.Ö. and Ö.Ö.; methodology, A.G., I.Ö. and Ö.Ö.; validation, A.G. and I.Ö.; formal analysis, I.Ö. and Ö.Ö.; investigation, A.G.; resources, A.G. and Ö.Ö.; data curation, A.G.; writing—original draft preparation, I.Ö. and Ö.Ö.; writing—review and editing, A.G., I.Ö. and Ö.Ö.; visualization, I.Ö. and Ö.Ö.; supervision, A.G.; project administration, A.G., I.Ö. and Ö.Ö. All authors have read and agreed to the published version of the manuscript.

**Funding:** This research received no external funding.

**Institutional Review Board Statement:** Not applicable.

**Informed Consent Statement:** Not applicable.

**Data Availability Statement:** Publicly available datasets were analyzed in this study.

**Conflicts of Interest:** The authors declare no conflict of interest.

## Appendix A. Variable Descriptions and Sources

The table provides descriptions and sources of the variables used in this study.

**Table A1.** Variable Descriptions and Sources.

| Variables | Description (Source) |
|---|---|
| State restrictions | First principal component of stay home orders, traveler quarantine, business closures, gathering bans, and postponement of elections (Author calculations; Kaiser Family Foundation, KFF). |
| Stay home | An index that equals 0 for states with no action, 1 for states with some action, 2 for states with stay home orders for high-risk groups, 3 for statewide stay home orders. The index is scaled to a range between 0 and 1 (Kaiser Family Foundation, KFF). |
| Traveler quarantine | An index that equals 0 for states with no action, 1 for states with some action, 2 for states with travel restrictions from certain states, 3 for states with restrictions on air travelers only, 4 for states with restrictions on all travelers. The index is scaled to a range between 0 and 1 (Author calculations; Kaiser Family Foundation, KFF). |
| Business closures | An index that equals 0 for states with no action, 1 for states with some action, 2 for states with non-essential business closures. The index is scaled to a range between 0 and 1 (Author calculations; Kaiser Family Foundation, KFF). |
| Gathering bans | An index that equals 0 for states with no action, 1 for states with some action, 2 for states prohibiting gatherings of more than 10 people, and 3 for states prohibiting all gatherings (Author calculations; Kaiser Family Foundation, KFF). |
| Postpone elections | An indicator that equals 1 for states with primary election postponement (Author calculations; Kaiser Family Foundation, KFF). |
| Case rate per 100,000 people | Confirmed COVID-19 cases per 100,000 people (New York Times COVID-19 repository). |
| Death rate per 100,000 people | Confirmed COVID-19 deaths per 100,000 people (New York Times COVID-19 repository). |
| Change in cases (%) | Percentage change in cases from the first quarter of 2020 to the second quarter of 2020 (Author calculations; New York Times COVID-19 repository). |
| Change in deaths (%) | Percentage change in deaths from the first quarter of 2020 to the second quarter of 2020 (Author calculations; New York Times COVID-19 repository). |
| High cases | A binary variable that equals 1 for the states with an above-median case rate per 100,000 people from COVID-19. |
| High deaths | A binary variable that equals 1 for the states with an above-median death rate per 100,000 people from COVID-19. |

**Table A1.** *Cont.*

| Variables | Description (Source) |
|---|---|
| Change in real GDP (%) | Percent change in state-level real GDP from the last quarter of 2019 to the first quarter of 2020 (Author calculations; Bureau of Economic Analysis). |
| Change in state coincident index (%) | Percent change in the state-level coincident index from the last quarter of 2019 to the first quarter of 2020. The state coincident index combines four state-level indicators of economic conditions: nonfarm payroll employment, average hours worked in manufacturing by production workers, the unemployment rate, and wage and salary disbursements plus proprietors' income deflated by the consumer price index (Author calculations; Federal Reserve Bank of Philadelphia). |
| Change in business earnings (%) | Percent change in net revenue for all businesses seasonally adjusted and measured at the end of the first quarter of 2020, relative to 4–31 January 2020 (The Opportunity Insights Economic Tracker: Chetty et al. 2020). |
| Change in small business earnings (%) | Percent change in net revenue for small businesses measured at the end of the first quarter of 2020, seasonally adjusted and relative to 4–31 January 2020 (The Opportunity Insights Economic Tracker: Chetty et al. 2020). |
| Change in business employment (%) | Percent change in employment for low-income workers in all businesses measured at the end of the first quarter of 2020 relative to 4–31 January 2020 (The Opportunity Insights Economic Tracker: Chetty et al. 2020). |
| Change in small business employment (%) | Percent change in employment for low-income workers in small businesses measured at the end of the first quarter of 2020 relative to 4–31 January 2020 (The Opportunity Insights Economic Tracker: Chetty et al. 2020). |
| Change in unemployment (%) | Percent change in state-level unemployment total insurance claims per 100 people in the labor force from the last quarter of 2019 to the first quarter of 2020 (Author calculations; Quarterly Business Dynamics Statistics, Department of Labor). |
| Change in bankruptcy filings (%) | Percent change in state-level business and personal bankruptcies from the first to second quarter of 2020 (American Bankruptcy Institute). |
| Change in retail and recreation mobility (%) | Percent change in mobility measured as of the end of the first quarter of 2020 relative to the five-week period of Jan 3-Feb 6 for places like restaurants, cafes, shopping centers, theme parks, museums, libraries, and movie theaters (Google Community Mobility Reports). |
| Change in groceries and pharmacy mobility (%) | Percent change in mobility measured as of the end of the first quarter of 2020 relative to the five-week period of Jan 3-Feb 6 for places like grocery markets, food warehouses, farmers markets, specialty food shops, drug stores, and pharmacies (Google Community Mobility Reports). |
| Change in residential mobility (%) | Percent change in mobility measured as of the end of the first quarter of 2020 relative to the five-week period of Jan 3-Feb 6 for places of residence (Google Community Mobility Reports). |
| Change in mobility in the parks (%) | Percent change in mobility measured as of the end of the first quarter of 2020 relative to the five-week period of Jan 3-Feb 6 for places like parks, national parks, public beaches, marinas, dog parks, plazas, and public gardens (Google Community Mobility Reports). |
| Change in transit mobility (%) | Percent change in mobility measured as of the end of the first quarter of 2020 relative to the five-week period of Jan 3-Feb 6 for places like public transport hubs such as subway, bus, and train stations (Google Community Mobility Reports). |
| Change in workplace mobility (%) | Percent change in mobility measured as of the end of the first quarter of 2020 relative to the five-week period of Jan 3-Feb 6 for places of work (Google Community Mobility Reports). |
| Change in overall mobility (%) | First principal component of changes in retail and recreation, groceries and pharmacy, park, transit, workplace, and residential mobility measured as of the end of the first quarter of 2020 relative to the five-week period of Jan 3-Feb 6 (Google Community Mobility Reports). |
| Pneumonia shot share for age 65 and over | Share of adults aged 65 and over who have ever received a pneumonia vaccine (Author calculations; Influenza and Pneumonia Deaths and Vaccinations, Kaiser Family Foundation, KFF). |
| Hospital beds per population | Hospital beds per 10,000 population (Health Care Provider Capacity, Kaiser Family Foundation, KFF). |
| Tests per population | Confirmed COVID-19 tests per 100,000 people (COVID Tracking Project). |
| Lagged change in real GDP | Percent change in state-level real GDP from the third quarter of 2019 to the fourth quarter of 2019 (Author calculations; Bureau of Economic Analysis). |
| State population | The population of the state in 2019 (Census Bureau). |

## Appendix B. Tests of Parallel Trends

The table reports test statistics for the assumption that treated and control group firms exhibited parallel trends, that is, the outcome variable, *Real GDP Growth*, manifested similar pre-shock trends. We reported the average *Real GDP Growth* for the treated and control groups, along with the tests of statistical differences in the three quarters before

the treatment (the "pandemic"). We defined two sets of treated groups based on the median value of the disease severity variables in the state-level sample using cases or deaths. Cases categorize countries as treated when they have above-median values of COVID-19 confirmed cases per million and categorize them as the control group otherwise. Deaths categorize countries as treated when they have above-median values of COVID-19 confirmed deaths per million and categorize them as the control group otherwise. Table 1 provides variable definitions. The *p*-values are reported beneath the coefficient estimates. *, **, and *** indicate significance at the 10%, 5%, and 1% levels, respectively.

**Table A2.** Test of Parallel Trends.

| Variables | Mean Control | Mean Treated | Difference | \|t\| | Pr(\|T\| > \|t\|) |
|---|---|---|---|---|---|
| **Panel A. Cases** | | | | | |
| *Real GDP Growth* | 1.84 | 2.08 | 0.24 | 1.10 | 0.28 |
| **Panel B. Deaths** | | | | | |
| *Real GDP Growth* | 1.93 | 1.98 | 0.04 | 0.20 | 0.84 |

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
