# Peer review of "The Impact of COVID-19 and Its Policy Responses on Local Economy and Health Conditions"

_jrfm, doi:10.3390/jrfm14060233_

Round 1

Reviewer 1 Report

Dear authors, 

Thank you for sending the reverse version of your manuscript.   

You have done a good job improving the paper. 

I think the changes are satisfactory and the paper can be accepted with minor language editing. 

Regards, 

Reviewer

Author Response

We would like to thank the reviewer for the valuable time and effort spent on evaluating the paper and providing constructive comments.

In this revision, we have reviewed and edited our language and style. We have also moved all the important footnotes to the main text. We have deleted two footnotes (footnotes 8 and 11) that referred to unreported results that were not central to our findings and conclusions. All revisions made to the manuscript are marked up using the “Track Changes” function, such that changes can be easily viewed by the editors and reviewers. Please see the attachment for the marked manuscript.

We believe that your thoughtful comments and our responses to them have substantially improved the paper. We hope that you find the current version of the manuscript ready for publication in the Journal of Risk and Financial Management.

Reviewer 2 Report

The authors dealt well with the several concerns raised by te reviewers. I consider that the paper is in conditions to be accepted.

Author Response

We would like to thank the reviewer for the valuable time and effort spent on evaluating the paper and providing constructive comments.

In this revision, we have reviewed and edited our language and style. We have also moved all the important footnotes to the main text. We have deleted two footnotes (footnotes 8 and 11) that referred to unreported results that were not central to our findings and conclusions. All revisions made to the manuscript are marked up using the “Track Changes” function, such that changes can be easily viewed by the editors and reviewers. Please see the attachment for the marked manuscript.

We believe that your thoughtful comments and our responses to them have substantially improved the paper. We hope that you find the current version of the manuscript ready for publication in the Journal of Risk and Financial Management.

This manuscript is a resubmission of an earlier submission. The following is a list of the peer review reports and author responses from that submission.

Round 1

Reviewer 1 Report

The manuscript 'The Impact of COVID-19 and its Policy Responses on Local Economy and Health Conditions' is well written and organized, and the topic is very relevant in today's world, especially in the USA where the Covid-19 pandemic has been severe. The results are compressive and sound, which suggests that the author(s) understand(s) the subject of interest.

Reviewer 2 Report

The author(s) assessed the policy interventions introduced during the COVID19 pandemic and concluded that the economic costs of lockdowns outweigh the economic damage of the disease.

My thoughts are as follows.

  • The Abstract and the introduction needs to remotivated.
  • Lin 22-25 is categorical statement that require citation.
  • I am curious as to why the author didn't include any citations in the introduction. This demonstrates that the research has no theoretical basis or empirical data.

  • There are millions of papers on policy responses, and the author sees no need to include a literature review section, which is a source of concern.

  • The manuscript did not mention any specific hypothesis formulations or tests.

  • There was no mention of the study's scope or the dataset's time span.

  • It's unclear where actual income data came from or why it was included in the manuscript.

  • The study has a technical flaw in that it does not account for quadratic or interaction terms. Ignoring this factor leads to inconclusive findings.

  • There is a bias due to omitted variables. Data on hospitalization rates is critical for successful policy during a pandemic.

  • According to the author, “more extreme prevention methods to control the disease were adopted through greater exposure to the disease.” This argument was not supported by the data or the findings.

Based on the aforementioned study bottleneck, I conclude that the paper needs substantial revisions before it can be accepted for publication.

Reviewer 3 Report

In this paper, the authors propose to analyse the impact of the COVID-19 in economy and health conditions in the US. The topic is relevant, but I have some doubts about methodological issues, which I believe that the authors do not deal correctly. 

  1. For example, it is not understandable why the authors choose some variables and not others. See, for example, Table 1. Why these variables?
  2. The variables of Table 1 are presented without any explanation. This explanation just appears in the Appendix A (which is not referenced, for example).
  3. Authors use a wide set of variables and propose to make regressions, but the way as the data is organized deserves, probably, panel data. Authors never talk about this possibility. Did they analyse that possibility? The use of regressions instead panel data, in some conditions, imply wrong results and conclusions.
  4. Moreover, using time series, authors should analyse the stationarity of the data, and this is not made.
  5. The presentation of the results is meaningless: tables and more tables, but without a relevant explanation of the results.
  6. I stayed somehow confused with some trasnformation of variables, as explained in appendix A. For example, "Stay home", authors have an index from 1 to 3 but then rescale the variable. Why

I believe that despite the intention of the authors, the paper fails totally in clearity, turning difficult the readibility of the paper and its correctness.